# 3D Morphable Master Face Generation: Towards Controllable Wolf Attacks against 2D and 3D Face Recognition Systems

## Abstract

Biometric authentication systems are facing increasing threats from Artificial Intelligence-Generated Content (AIGC). Previous research has revealed the vulnerability of face authentication systems against master face attacks. These attacks utilize generative models to create facial samples capable of matching multiple registered user templates in the database. In this paper, we present a systematic approach for generating master faces that can compromise both 2D and 3D face recognition systems. Notably, our approach is the first to enable morphable and controllable master face attacks on face authentication systems.

Our method generates these 3D master faces using the Latent Variable Evolution (LVE) algorithm with the 3D Face Morphable Model (3DMM). Through comprehensive simulations of simultaneous master face attacks in both white-box, gray-box, and black-box scenarios, we demonstrate the significant threat posed by these 3D master faces to mainstream face authentication systems. Furthermore, we explore the realms of face morphing and facial reenactment in our generated samples, enhancing the efficacy of the master face attack. Compared to existing methods, our approach exhibits superior attack success rates and advanced flexibility, highlighting the importance of defending against master face attacks.

## 1 Introduction

Identity-based security against cyber-attacks has long been a concern, but recent developments in Artificial Intelligence-Generated Content (AIGC) have brought renewed attention to cybersecurity, particularly in relation to biometric authentication systems. These AIGC techniques can be abused to create fake biometric traits, a notable example being their manipulation in deepfakes to fool face recognition (FR) systems.

While deepfake techniques typically require access to a victim's biometric information, another attack method, known as the "**wolf attack**" introduced by Une et al. (2007), enables attackers to execute unconditional attacks by generating generic "**master samples**" that closely resemble multiple enrolled templates within an FR system's gallery. Nguyen et al. (2020) demonstrated the generation of master face samples without the need for specific victim information. These face samples effectively cover a wide range of identities in the gallery, exposing the vulnerability of face-based authentication systems to this master attack.

However, previous studies on master face attacks have predominantly focused on 2D scenarios. Despite their widespread use, 2D FR systems often experience decreased accuracy under challenging conditions such as changing poses, facial expressions, illumination variations, and occlusions (Zhou & Xiao, 2018). Moreover, their reliance solely on facial appearance makes them vulnerable to spoofing attacks. In contrast, 3D FR systems that leverage additional depth information generally exhibit greater robustness in demanding scenarios and higher resistance to common spoofing attacks, such as presentation attacks. With the availability of affordable commercial 3D sensors, 3D FR systems have become mainstream, prompting the need to further investigate the applicability of master faces in the 3D domain.

The first attempt to generate 3D master faces was introduced by Friedlander et al. (2022). Their methodology relies mainly on the StyleGAN2 network to generate 2D facial samples, followed by

the reconstruction of 3D facial geometries from these images. While this approach successfully produces usable 3D master face samples, it exhibits notable limitations. The latent space of StyleGAN2 was not originally designed to preserve explicit 3D information, potentially resulting in the loss of critical 3D facial characteristics learned from training data and 3D FR systems. Additionally, the reconstruction process typically introduces errors, affecting the fidelity of the generated 3D faces. Furthermore, precise attribute control, especially for facial pose and expression, remains challenging for StyleGAN2, limiting the applicability of the generated master faces for spoofing.

In this paper, we introduce a novel framework that employs a latent variable evolution (LVE) strategy with a Face Learning and Expression (FLAME) generator, a widely used 3D Morphable Face Model (3DMM), to generate 3D master face samples. The LVE strategy progressively improves the impersonation success rate of our generated samples through an iterative optimization process guided by pre-trained 2D and 3D FR systems. FLAME, as presented by Li et al. (2017), is designed to disentangle shape, appearance, expression, and pose parameters, allowing the production of anatomically plausible and highly controllable 3D facial shapes. This attribute disentanglement better preserves critical 3D information, resulting in improved controllability and threat potential.

Additionally, we present the first evaluation of master face sample generalizability against 2D and 3D FR systems using a variety of 3D face datasets, including FaceScape (Yang et al., 2020), BU-3DFE (Yin et al., 2006), Headspace (Dai et al., 2020), and Texas3D (Gupta et al., 2010) by simulating master face attacks in white-box, gray-box, and black-box attack scenarios. We also explore the practical application of reenactment and morphing of 3D master faces, demonstrating how these morphable 3D master faces can significantly enhance master face attacks, highlighting the importance of strengthening FR systems to defend against such attacks.

In summary, our contributions in this paper are as follows: 1) We propose a straightforward yet efficient method that significantly enhances the threat and usability of 3D master faces. 2) We perform the first extensive evaluations across diverse 3D face datasets and various FR systems, encompassing both white-box, gray-box, and black-box attack scenarios to simulate real-world master face attacks. 3) We demonstrate how a controllable master face can further enhance potential attacks through facial reenactment and morphing. Our work emphasizes the need for future research on FR systems to be aware of the threat from master face attacks and take corresponding defensive countermeasures.

## 2 RELATED WORK

**Face Recognition System**   The last decade has seen the rapid development of deep learning methods for 2D face recognition. A significant milestone was the introduction of DeepFace by Taigman et al. (2014), which achieved an impressive accuracy rate of 97.35% on the LFW benchmark (Huang et al., 2008), approaching human-level performance. Subsequently, the application of convolutional neural networks (CNNs) in FR systems flourished. Schroff et al. (2015) presented FaceNet, which was trained with a triplet loss function on a GoogleNet architecture. Different from the traditional Euclidean distance-based loss, Liu et al. (2017) proposed a novel angular softmax loss. Wang et al. (2018) and Deng et al. (2019a) further introduced additive cosine/angular margin to tackle optimization challenges for the aforementioned loss function. Recent research has explored adaptive loss functions, including the adaptive margin for image quality introduced by Kim et al. (2022).

In contrast, 3D FR systems, known for their superior performance in challenging cases compared to their 2D counterparts, have received less attention in deep learning-based research. This is partly due to the scarcity and privacy sensitivity of 3D facial training data. The first deep convolutional network-based 3D FR model, introduced by Kim et al. (2017), involved fine-tuning the pre-trained 2D VGGFace model (Cao et al., 2018) with facial depth maps. Gilani & Mian (2018) combined publicly available 3D face datasets to create a comprehensive dataset and trained a CNN-based model from scratch called FR3DNet. To address the challenges posed by the lack of high-quality training data, Mu et al. (2019) proposed Led3D, an open-source lightweight CNN model that uses low-quality depth images for training.

**Face Generation**   Among the various generative models for creating 2D facial images, the framework of Generative Adversarial Networks (GANs), originally proposed by Goodfellow et al. (2014), is noteworthy. GAN can be conceptualized as a two-player minimax game between the Generator and the Discriminator. The Generator is a differentiable function that transforms an initial latent

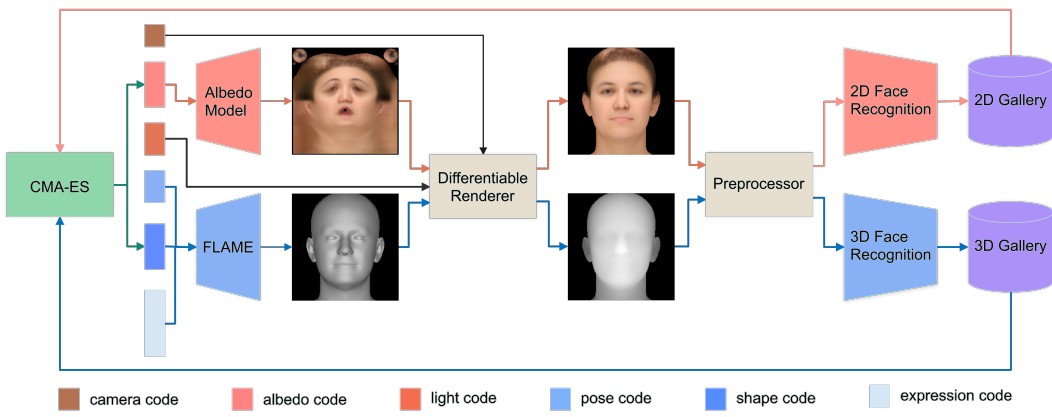

Figure 1: The Latent Variable Evolution process with 3D Morphable Face Model

vector into a data sample, striving to generate data that closely resembles real training data. In contrast, the Discriminator is trained to differentiate between samples generated by the Generator and real training data. A significant development is StyleGAN, which was introduced by Karras et al. (2019). This variant includes a mapping network that separates content and style information, leading to improved control over the appearance of generated images.

Our research emphasizes 3D face generation methods, particularly those involving the widely used 3D Morphable Face Model (3DMM) introduced by Blanz & Vetter (1999). This model disentangles facial components such as shape, appearance, and expressions, facilitating statistical capture of variations and tasks like facial reenactment. The preprocessing stage establishes point-to-point correspondence within the training databases, which enables meaningful combinations of faces and allows face generation through proper coefficient sampling(Egger et al., 2020). Furthermore, Analysis-by-Synthesis techniques can be used to estimate these coefficients from 2D images, which is a prominent approach in modern 3D face reconstruction models, including DECA proposed by Feng et al. (2021).

**Master Attack**   The Wolf Attack, also known as the Master Attack, was introduced by Une et al. (2007). This attack aims to create a generic sample capable of falsely matching multiple enrolled subjects in a biometric authentication system's gallery. Initially applied to fingerprint-based authentication systems by Bontrager et al. (2018), this concept was extended to face-based authentication systems by Nguyen et al. (2020). Recent research by Shmelkin et al. (2021) and Nguyen et al. (2022) analyzed master faces, exploring their properties and assessing their generalizability across different datasets and 2D FR systems. Our work advances the field by generating 3D master faces that can effectively target both 2D and 3D face recognition systems concurrently. We conduct experiments involving various scenarios to replicate real-world master attacks' practical applicability.

## 3   PROPOSED METHOD

### 3.1   OVERVIEW

Our training process for 3D master face generation is demonstrated in Figure 1. Building upon the concept of a "wolf sample"(Une et al., 2007), we denote our collection of authentic human templates as $\mathbb{T}_h$. While numerous publicly available 3D facial datasets primarily consist of human face meshes, referred to as $\mathbb{S}_h$, many 3D FR systems use depth images as input rather than the entire mesh. To accommodate this, we developed a data preprocessing pipeline outlined in the Appendix A.1, labeled as $P$. This pipeline transforms $\mathbb{T}_h$ into RGB-D image pairs for each facial scan, resulting in $\mathbb{T}_h = \{t_h \mid (c_h \in \mathbb{R}^{w_c \times h_c \times 3},\ d_h \in \mathbb{R}^{w_d \times h_d})\}$, where $\mathbb{C}_h$ and $\mathbb{D}_h$ represent sets of color and depth images, respectively.

Face authentication systems employ FR models to encode input images to lower-dimensional feature representations. For 2D FR, the function $f_{2d} : \mathbb{R}^{w_c \times h_c \times 3} \rightarrow \mathbb{R}^d$ maps color images to a

$d$-dimensional space. A similar function is employed for 3D FR, utilizing depth images as input instead, resulting in potentially distinct intermediate dimensionalities.

The face matching function $m : \mathbb{R}^d \times \mathbb{R}^d \to \{0, 1\}$ is used to predict whether the embeddings of the two inputs correspond to the same identity. This matching function $m$ is conditioned on a chosen threshold $\theta$, specific to the selected similarity metric, in our case, the cosine similarity metric between feature embeddings. However, our work necessitated the simultaneous consideration of RGB-D matching, leading to a more complex matching function $M$:

$$M(\boldsymbol{t}_1, \boldsymbol{t}_2, \theta_{\text{2d}}, \theta_{\text{3d}}) = m_{\text{2d}}(f_{\text{2d}}(\boldsymbol{t}_{1_c}), f_{\text{2d}}(\boldsymbol{t}_{2_c}), \theta_{\text{2d}}) \cap m_{\text{3d}}(f_{\text{3d}}(\boldsymbol{t}_{1_d}), f_{\text{3d}}(\boldsymbol{t}_{2_d}), \theta_{\text{3d}})$$

Based on the above notation, our objective in master face generation is to produce a forged sample $\mathbf{x}$ that can match the highest number of enrolled templates:

$$\boldsymbol{x} = \arg\max_{\boldsymbol{x}} \sum_{\boldsymbol{t} \in \mathbb{T}_h} M(\boldsymbol{x}, \boldsymbol{t}, \theta_{\text{2d}}, \theta_{\text{3d}})$$

To achieve this goal, we employ a 3DMM-based face generator $G$ to synthesize a 3D face mesh $\boldsymbol{s}$. This synthesis is conditioned on a set of latent codes $\boldsymbol{c}, \boldsymbol{\alpha}, \boldsymbol{l}, \boldsymbol{\beta}, \boldsymbol{\theta}, \boldsymbol{\psi}$, corresponding to camera, albedo, light, shape, and expression (Li et al., 2017). To simplify the training procedure, we freeze other latent codes and only optimize the albedo and shape codes during training. We then utilize the same data preprocessor $P$ to produce the RGB-D image pair of this synthesized face. With our 3DMM generator, the master face generation problem can now be formulated as finding an optimal pair of latent vectors $(\boldsymbol{\alpha}, \boldsymbol{\beta})$ that results in the highest false matching rate (FMR):

$$(\boldsymbol{\alpha}, \boldsymbol{\beta})_{opt} = \arg\max_{(\boldsymbol{\alpha}, \boldsymbol{\beta})} \frac{\sum_{\boldsymbol{t} \in \mathbb{T}_h} M(P(G(\boldsymbol{\alpha}, \boldsymbol{\beta})), \boldsymbol{t}, \theta_{\text{2d}}, \theta_{\text{3d}})}{\|\mathbb{T}_h\|}$$

In summary, our approach first transforms the 3D facial dataset to the template database $\mathbb{T}_h$ with RGB-D image pairs, from which we extract their features using 2D and 3D FR systems. Subsequently, we insert a pair of random albedo and shape vectors into the 3D face generator $\mathbf{G}$ and preprocess the resulting synthesized face to obtain an RGB-D image pair, which is further encoded and verified alongside other feature embeddings from $\mathbb{T}_h$ to determine the count of successful matches.

Maximizing the count of matches necessitates an iterative process to refine $(\boldsymbol{\alpha}, \boldsymbol{\beta})$. For this purpose, we introduce a Latent Variable Evolution strategy in the following Section 3.2.

### 3.2 LATENT VARIABLE EVOLUTION ALGORITHM

We formalized the process for refining an initial latent vector $(\boldsymbol{\alpha}, \boldsymbol{\beta})$ as outlined in Algorithm 1. To address optimization challenges inherent in our master face generation problems, which involve non-differentiable components(such as cosine similarity calculation and thresholding), we employed the Covariance Matrix Adaptation Evolution Strategy (CMA-ES) (Hansen et al., 2019) as our optimizer.

Our implementation of the LVE algorithm leverages the ask-and-tell interface of CMA-ES. First, we initialize the CMA-ES solver with random latent codes. When we "ask" the solver for solutions, it generates potential candidate solutions by sampling from a multivariate normal distribution with parameters determined by the initialization. We execute the complete generation and matching procedure using these candidate solutions, to obtain fitness scores from our objective function. These fitness scores are subsequently "told" to the CMA-ES optimizer. The optimizer utilizes this feedback to update its distribution parameters, including the mean vector and covariance matrix, for the subsequent iterations of the ask-and-tell process. This iterative approach enables the optimizer to progressively explore the search space, ultimately converging towards an optimal solution.

One of the key challenges of the LVE algorithm is finding an effective initialization point. If the initialization is significantly distant from the optimal solutions, it can slow down convergence or trap the optimizer in local optima. Fortunately, our method benefits from the nature of the statistical 3DMM, that the zero vectors represent the mean appearance and shape computed from a large 3D face dataset, making the zero vectors excellent starting points.

Another challenge lies in defining an appropriate objective function that guides the CMA-ES algorithm effectively toward improved solutions. In prior studies on 2D master face generation proposed

---

**Algorithm 1** Latent Variable Evolution pseudo code

---

$m \leftarrow 22$               ▷ Population size
$\mathbb{F} \leftarrow \{\}$               ▷ Master face set
$\mathbb{S} \leftarrow \{\}$               ▷ Score vector
$(\boldsymbol{\alpha}, \boldsymbol{\beta}) \leftarrow \mathbf{0}$               ▷ Initialization
**for** $n$ iterations **do**               ▷ Run LVE for $n$ times
    $\mathbb{X} \leftarrow P(G((\boldsymbol{\alpha}, \boldsymbol{\beta})))$          ▷ Generate $m$ faces
    $\boldsymbol{s} \leftarrow \mathbf{0}$          ▷ Initialize scores $\boldsymbol{s} \in \mathbb{R}^m$
    **for** face $\boldsymbol{x}_i$ in faces $\mathbb{X}$ **do**
        **for** face $\boldsymbol{t}_j$ in faces $\mathbb{T}$ **do**
            $s_i \leftarrow s_i + M(\boldsymbol{x}_i, \boldsymbol{t}_j, \boldsymbol{\theta})$      ▷ $\boldsymbol{\theta}$ is increasing progressively
        **end for**
    **end for**
    $\boldsymbol{s} \leftarrow \frac{\boldsymbol{s}}{|\mathbb{T}|}$          ▷ Calcuating FMR
    $\boldsymbol{x}_b, \boldsymbol{s}_b \leftarrow \text{GetBestFace}(\mathbb{X}, \boldsymbol{s}, \boldsymbol{\theta})$
    $\mathbb{F} \leftarrow \mathbb{F} \cup \{\boldsymbol{x}_b\}$          ▷ Append best master face
    $\mathbb{S} \leftarrow \mathbb{S} \cup \{\boldsymbol{s}_b\}$          ▷ Append best score
    $(\boldsymbol{\alpha}, \boldsymbol{\beta}) \leftarrow \text{CMA-ES}(\mathbf{1} - \boldsymbol{s})$
**end for**
**return** $\mathbb{F}, \mathbb{S}$
$\boldsymbol{f}_b, \boldsymbol{s}_b \leftarrow \text{GetBestFace}(\mathbb{F}, \mathbb{S}, \boldsymbol{\theta})$      ▷ Find master faces under the final threshold $\boldsymbol{\theta}$

---

by Nguyen et al. (2020), the optimization process relied on similarity scores between two faces, with the aim of increasing these scores. However, our work introduces complexity by incorporating both 2D and 3D FR, emphasizing simultaneous matches. Enhancing similarity scores for both 2D and 3D FR might not yield the desired outcomes in our context, as shown in the Appendix A.3.1. This is because a face sample with high average similarity scores in both 2D and 3D FR could be matched to different individuals across modalities due to distinct feature space distributions. To address this challenge, we rely on the matching function described in 3.1, which quantifies the count of concurrent matches for the same individual in both 2D and 3D scenarios.

Our chosen objective function introduces a potential challenge related to the low FMR from pre-trained FR systems. Mature pre-trained FR models can result in extremely low matching numbers or even zero matches, leading to meaningless fitness scores for the CMA-ES solver, especially during the initial stages of optimization when the initial guess is far from the optima. To overcome these optimization difficulties, we adopt a strategy to initialize the threshold $\boldsymbol{\theta}$ with relatively small values. This initial choice ensures a high FMR, thereby promoting meaningful fitness scores for the optimization process. As we progress through optimization, we gradually increment the threshold value, compelling the latent vector to shift towards the desired optima in the feature space.

### 3.3 RECONSTRUCTION-BASED BASELINE

From the problem statement above, it becomes evident that selecting an appropriate 3D face generator, denoted as $G$, plays a key role in ensuring the performance and efficiency of the method. In the study conducted by Friedlander et al. (2022), a reconstruction-based generator was employed to reconstruct the 3D geometry from images generated through StyleGAN2 (Deng et al., 2019b). Consequently, during the LVE optimization process, the latent vector was manipulated within a 2D context, rather than directly in the 3D domain, which introduced several limitations.

Firstly, the instability of the StyleGAN2 generator presented a challenge. Randomly sampling the latent vector could yield human faces with varying poses and expressions. Faces with unfavorable expressions or poses proved difficult to optimize, ultimately leading to decreased performance. To address this issue, Friedlander et al. (2022) ran the LVE algorithm five times and selected the optimal outcome for evaluation. Although effective, this method is computationally expensive.

Furthermore, optimizing within the 2D domain during the optimization stage compromised the information available from 3D FR. 3D face reconstruction from a single image is an ill-posed problem. Therefore, the reconstruction process typically introduces inaccuracies and uncertainties, leading to

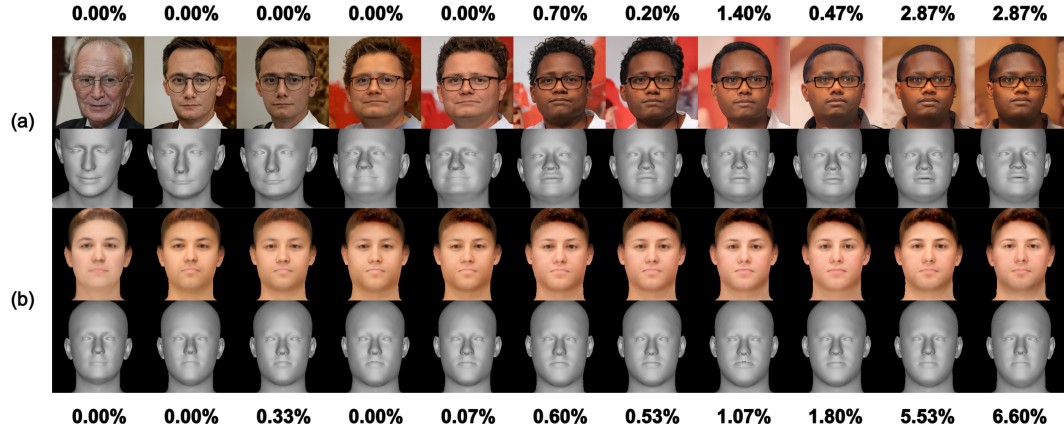

Figure 2: The intermediate faces and their **joint FMRs** on the training set. Row (a) is generated by the baseline and row (b) is generated by our method. The leftmost column is the initialized face sample and the rightmost is the master face sample we get after 1,000 iterations.

Table 1: Equal Error Rates (%) computed on each Dataset-FR System pair

|          | FaceNet | AdaFace | IResnet100 | Led3D |
|----------|---------|---------|------------|-------|
| BU-3DFE  | 1.17    | 10.35   | 9.27       | 11.70 |
| Texas3D  | 0.08    | 6.64    | 4.69       | 4.00  |
| Headspace| 1.95    | 1.70    | 1.79       | 1.29  |

a loss of information related to the characteristics of the 3D master face. Additionally, since the 3D geometry is estimated from 2D images, controlling the 3D domain without affecting 2D appearance is challenging, resulting in reduced controllability.

We highly appreciate the design principles embedded in 3DMMs, particularly their ability to disentangle appearance, shape, and expression models, enabling flexible control. In our proposed methodology, we manipulated the shape vector to govern the generated geometry, while the appearance parameter controlled visual attributes. These latent codes could be optimized jointly or independently by CMA-ES solvers based on the feedback from 2D and 3D FR, allowing more effective utilization of information from both modalities.

For comparison, as shown in Figure 2, we re-implemented the reconstruction-based method using StyleGAN2 and DECA, rather than using the original reconstruction network introduced by Deng et al. (2019b). The reason is that DECA employed the FLAME topology for 3D face reconstruction as well, enhancing fairness in comparisons. Furthermore, DECA achieved better reconstruction performance than the work mentioned above over the NoW benchmark (Sanyal et al., 2019). As demonstrated in the forthcoming section, our disentangled approach not only yielded improved FMR but also increased controllability, allowing for easier reenactment and morphing.

## 4 EXPERIMENT

### 4.1 DATASET AND FACE RECOGNITION SYSTEM

In our experiments, we utilized four 3D face datasets and four different FR systems, allowing us to explore various configurations and assess the generalizability of the master faces. To evaluate the performance of the FR systems, we measured their Equal Error Rates (EER) in Table 1. For a more comprehensive introduction to the datasets and FR systems used, please refer to Appendix 2.

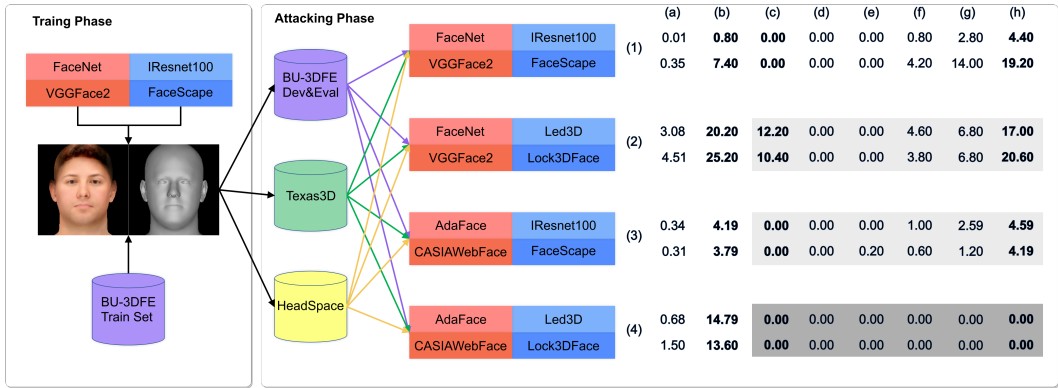

Figure 3: Master face attack scenarios we evaluated and the partial results from Table 3. We present the **joint FMRs** over four of these settings: (1) BU-3DFE + (FaceNet, IResnet100); (2) BU-3DFE + (AdaFace, Led3D); (3) Headspace + (FaceNet, IResnet100); (4) Texas3D + (AdaFace, Led3D), with the color from white to dark gray denoted white-box, gray-box and black-box attack. The unit of the data cell is the percentage (%).

Each row is divided into upper results for the dev set and lower parts for the eval set. The columns refer to joint FMRs of (a) natural faces' average FMRs on the test set; (b) natural master face on the test set; (c) natural master face on the train set; (d) baseline master face; (e) three baseline master faces generated in greedy strategy; (f) our master face; (g) our three master faces generated in greedy strategy; (h) (g) plus 27 morphs in between our master faces. Detailed results and analysis refer to Appendix A.2
.

## 4.2 MASTER FACE ATTACK SIMULATION

**Master Face Existence**  Nguyen et al. (2022) claimed that the master face phenomenon is due to the imbalanced distribution within the FR system. Deep learning-based FR systems often suffer from non-uniform distributions in the feature space. Consequently, if a face falls within a densely clustered region in the feature space, its likelihood of being falsely matched to other samples within that cluster increases. Training a master face can be regarded as approaching the densest cluster within the feature space of the FR system. However, these dense clusters may not align between two different systems, making it challenging to pinpoint cross-modal clusters of these vulnerable faces.

**Evaluation Anchor**  Given the absence of a benchmark for master face attacks evaluation, we selected two anchors as references to better analyze our results. In Figure 3, columns (a) to (c) present the results of natural master faces. (a) and (b) were computed under the experimental conditions corresponding to their respective rows in white-box conditions. Column (b) showcases the best achievable joint FMR that natural faces can attain when the target dataset and FR system are known. By comparing this with our results, we can assess the generality of our master faces.

The natural master faces shown in column (c), however, were computed under the same settings as our master face generation. We used these natural master faces from column (c) to perform attacks in other scenarios, equivalent to what we did with our master faces, namely "**natural master face attack**", with results noted in Table 4. Therefore, the results in column (c) cover not only white-box but also gray-box and black-box attacks. Since the master faces and the natural master faces from column (c) are computed/trained from the same source dataset and FR systems, their differing results intuitively evaluate the performance of our method.

**Single Master Face Generation**  We ran the LVE algorithm with 1,000 iterations on a BU-3DFE training set consisting of 1,500 facial data samples to train our master faces. The FR systems used in our experiments were FaceNet for 2D images and fine-tuned iResNet100 for 3D depth maps. Notably, the training set for the FR systems was distinct from the training set for the LVE algorithm.

To compare our 3DMM-based master face generation method with the reconstruction-based method, we ran the reconstruction-based method multiple times, each with a different initialization. We then

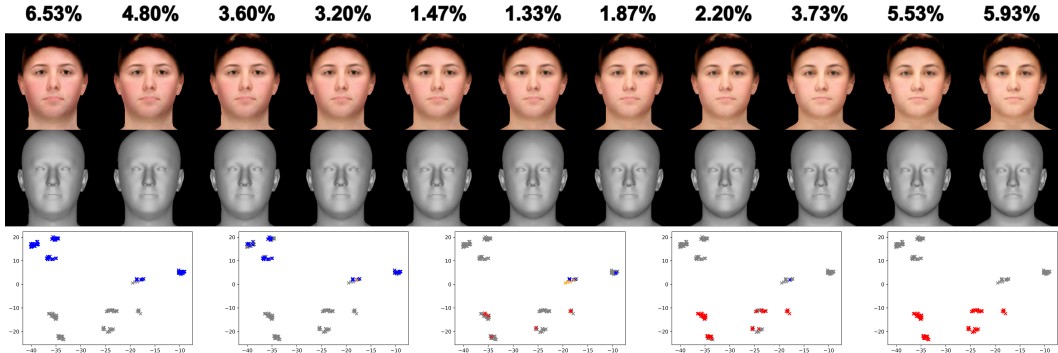

Figure 4: Effect of master face morphing. Columns show generated face samples with their **joint FMR** on top. From left to right, linear interpolation weight increases from 0.1 to 0.9. T-SNE visualization displays matching results for the left source master face, morph with weights 0.2, 0.5, 0.8, and right source master face, respectively. The orange points represent newly matched samples.

selected the result with a realistic visual appearance. As shown in Figure 2, the search space gradually approaches a virtual "cross-modal cluster" of faces vulnerable to master face attacks as the thresholds increase, leading to improved joint FMR on 2D and 3D FR systems. From observations, StyleGAN2 tightly entangles shape, appearance, head pose, and expression attributes, leading to joint adjustments during the optimization process. In contrast, our 3DMM disentangles these attributes, enabling optimization with fewer degrees of freedom and resulting in better FMR results.

**Master Face Attack**   The question of whether a master face generated from a training set can successfully generalize to a real-world face authentication system with unknown FR architectures or dataset distributions remains open. Generalization in this cross-modal scenario has proven challenging. Even in the simplest scenario of a white-box attack, the FMRs on the dev and test sets can be zero when attacking with only a single master face generated from the training set. To address this issue, we adopted a greedy strategy, which starts by generating one master face from the training set. Subsequently, individuals that have already been matched are removed, and another face is generated repeatedly. This strategy enables the exploration of more possible clusters of master faces in the feature space of the training set, with no overlap in individuals matched by each face. We use this set, rather than a single master face, to conduct the master face attack.

The experimental results depicted in Figure 3 illustrate that our methods, when combined with the greedy strategy, achieve higher FMRs than the best natural master faces and the baseline methods in a white-box attack scenario (1). In the gray-box attack scenario (2), where the FR systems' architecture is unknown, our methods fail to surpass the natural master faces, which are **computed under white-box conditions**, serving as a reference of "best natural results", while still exhibiting a significantly higher FMR than the average FMR. In gray-box attack scenario (3), where we lack knowledge about the dataset distribution, our methods maintain a relatively high FMR when the distribution is similar (as we use the same preprocessing pipeline for Headspace and BU-3DFE datasets). However, our master face attacks fail in the most demanding black-box attack scenario.

We have provided a comprehensive summary of our master face attack settings and analyses in Appendix A.2, which additionally includes an experiment involving natural master face attacks as depicted in column (c). We highly encourage readers to refer to Table 3 and 4 for detailed insights.

## 4.3   MASTER FACE MORPHING

While the greedy strategy has proven effective in improving the master face attack, it does come with a significant time cost when generating a larger number of master faces. The inherent nature of the LVE algorithm dictates that each training procedure results in just one master face sample. Running with 1,000 iterations, the baseline method takes approximately 14 hours to create a single master face, using an NVIDIA Tesla V100 card. Our 3DMM-based approach reduces this time cost by 1 hour as it omits the StyleGAN generation steps, but it still remains relatively high.

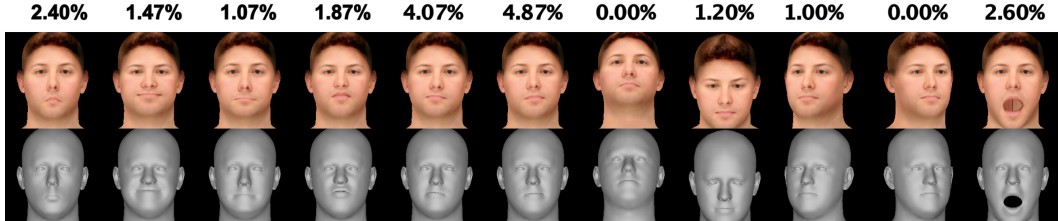

Figure 5: Effect of master face reenactment. Columns show generated face samples with their **joint FMR** on top. The first to sixth columns show the variations on the first three principal components of the expression. The other columns are visualizations to change the head pose and jaw articulation.

Our approach advances in swiftly generating new master faces through interpolation between existing master faces, supported by the exceptional interpolation control capabilities of 3DMMs. These morphs effectively preserve both shape and appearance, as shown in Figure 4. They cover a subset of mismatched identities from the source master faces and potentially introduce new mismatches.

Furthermore, we enhance the master face attack by incorporating these morphs. For example, creating 27 morphs from pairs of 3 master faces takes less than a minute. Employing these 30 samples in the attack significantly improves the attack success rate, as illustrated in Figure 3.

### 4.4 MASTER FACE REENACTMENT

While StyleGAN2 represents a significant advancement in generating high-quality images, it inherently lacks semantic control over the generated output. Consequently, tasks involving the precise control of expression and pose, while retaining identity information, remain challenging, leading to subsequent works such as Tewari et al. (2020). In contrast, our method, built upon the foundation of 3DMMs, offers significantly enhanced controllability, as presented in Figure 5.

Due to the sensitivity of 2D FR systems to pose variations, the success rate of attacks targeting specific poses may be relatively low. Nonetheless, our results still highlight the potential of utilizing a controllable 3D master face to strengthen presentation attacks against 2D face authentication systems, particularly against systems that require users to exhibit specific facial expressions.

## 5 DEFENSE AGAINST 3D MASTER FACE ATTACK

Our research has identified significant concerns regarding the vulnerability of 2D and 3D FR systems against controllable 3D master face attacks. Despite extensive research on security for 2D FR systems in the past decade, these findings do not seamlessly extend to 3D FR systems. For instance, presentation attack detection (Hernandez-Ortega et al., 2023) and deepfake detection (Rathgeb et al., 2022) can be readily adapted to counter physical and digital 2D morphing face attacks, respectively. However, similar work has not yet been achieved for 3D FR systems, which underscores the urgent need for research and development in this area. Another concern is the generalizability of detectors for both 2D and 3D FR systems, which remains an active research topic in biometric security.

## 6 CONCLUSION

We have introduced a novel master face attack method that leverages 3D Morphable Face Models to generate morphable and controllable master faces. As the first study to evaluate master face attacks against 2D and 3D FR systems across various attack scenarios, our greedy generation and morph creation method demonstrates the potential to compromise the face authentication systems, even when the architectures of FR systems or face gallery distributions are unknown. In addition, with disentangled parameters, we can easily change the facial expressions and poses of the master faces while retaining the ability of false matching. Based on these findings, we underscore significant security risks associated with controllable master face attacks and emphasize the need for further research in defense strategies.

## 7 ETHICS STATEMENT

The purpose of this research is to highlight the potential security risks associated with controllable 3D master face attacks and to encourage the development of more robust and secure biometric authentication systems. We strictly prohibit any abusive or illegal applications of our findings and advocate for the development of robust defense countermeasures against the method we have proposed. Additionally, we affirm that all 3D face datasets used in this work were obtained and applied in accordance with official ethical guidelines to prevent misuse or unauthorized applications.

## 8 REPRODUCIBILITY STATEMENT

To ensure reproducibility, we formulated the objective function for the CMA-ES solver in 3.1 and described the Latent Variable Algorithm in detail with pseudo-code in Algorithm 1. We listed the datasets and FR systems that we used for implementation in Appendix A.1. The code will be made publicly available after the paper decision has been made.

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

Table 2: Details of 3D facial datasets used in our experiments.

| Database | Year | Data type | IDs | Scans | Exps |
|----------|------|-----------|-----|-------|------|
| BU-3DFE | 2006 | Mesh | 100 | 2,500 | 25 |
| Texas3D | 2010 | Range Images | 118 | 1,149 | Various |
| Headspace | 2017 | Mesh | 1,519 | 1,519 | 1 |
| FaceScape | 2020 | Mesh | 847 | 16,940 | 20 |

## A APPENDIX

While our primary contributions and findings have been extensively discussed in the main content of this paper, this appendix serves as an additional resource to provide readers with a more comprehensive understanding of our experiments and conclusions. Here, we aim to provide the following supplementary information:

1. In Section A.1, we offer detailed introductions to the datasets and FR systems employed in our experiments. Additionally, we provide insights into how we evaluated the performance of these FR systems to ensure reproducibility and transparency.

2. While our paper presents partial results of master face attacks in Figure 3, this section presents the complete set of experimental results. We delve into the details of our white-box, gray-box, and black-box master face attack evaluations conducted on multiple datasets and FR systems in Table 3, compared with the natural master face attacks results in Table 4. Furthermore, we explore additional experiments targeting individual FR systems using our proposed method.

3. In Section A.3, we provide additional ablation studies, particularly focusing on the objective function used in our master face generation. We investigate how score-based and FMR-based objective functions impact master face generation differently. We also examine the necessity of incorporating regularization terms into the objective function and assess the impact of their respective weights.

4. At last, we discuss the limitations of our method and propose potential directions for future research in Section A.4.

### A.1 EXPERIMENT SETTING DETAILS

**Dataset** In Table 2, we provide comprehensive details about the 3D facial datasets utilized in our experiments. For the purpose of master face generation, we extracted 60 individuals, comprising a total of 1,500 scans, from the BU-3DFE dataset to form our training set. To maintain an extensive evaluation, the remaining 40 identities, similar to the Headspace and Texas3D datasets, were randomly shuffled and allocated to the development (dev) and evaluation (eval) sets. Specifically, the dev set of each dataset was used for conducting a grid search to identify an optimal threshold that effectively balances the False Acceptance Rate (FAR) and False Rejection Rate (FRR), ultimately minimizing the Equal Error Rate (EER), as outlined in Table 1. Notably, due to the unique characteristics of Headspace, with only one sample per individual, we manually selected thresholds to ensure that both 2D and 3D FR systems achieved an EER of less than 2% on the Headspace dev set.

Regarding FaceScape, although it has the largest number of samples, its facial topology does not include eyes and mouth, making it unsuitable for training the master face. We used the FaceScape dataset to generate 300 different individuals using the bilinear model, with each individual having 52 different expression meshes rendered in 9 different poses. Following the work by Kim et al. (2017), we used these rendered 3D depth maps to fine-tune a pre-trained 2D FR system, resulting in a workable 3D FR system.

**Face Recognition System** Among many research on 2D open-source FR systems, we selected FaceNet and AdaFace. FaceNet is based on the architecture of GoogleNet, also known as InceptionNet, being trained with the triplet loss. As a highly influential 2D FR model widely used to this day, FaceNet has demonstrated high efficiency and accuracy. We chose a model pre-trained on the VGGFace2 dataset for the experiments. In contrast, recent work by AdaFace introduced a novel loss

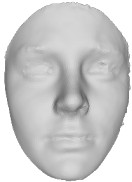 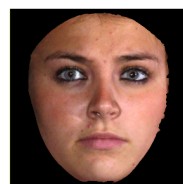 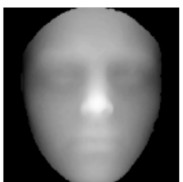 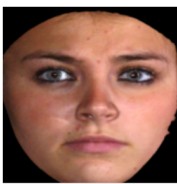

Figure 6: An example data of BU-3DFE. From left to right is the image of the origin 3D mesh, 2D face image, and pre-processed RGB-D images as the input for the FR systems. Note that images of this identity are provided as sample data illustrations on the BU-3DFE dataset's official website.

function based on adjustable image quality. We selected a model pre-trained on CASIA-WebFace(Yi et al., 2014) using ResNet18 as the backbone.

There are relatively limited open-source models on 3D FR systems, primarily due to the scarcity of open-source databases. Hence, as we mentioned above, we fine-tuned an IResnet100 which was originally trained on the MS1MV2 dataset(Deng et al., 2019a). Another 3D FR we utilized is a Led3D model constructed on a lightweight CNN architecture that incorporates a spatial attention vectorization module for multi-level feature fusion. Initially pre-trained on a combination of the FRGC v2(Phillips et al., 2005) and Bosphorus(Savran et al., 2008) datasets, it is further fine-tuned using the Lock3DFace(Zhang et al., 2016a) dataset, which consisted of Kinect-captured low-quality 3D face images. By carefully selecting the pre-trained 2D and 3D FR, we make sure that their training sets do not overlap with the dataset we used for training and evaluation.

**Data Preprocessing**   Our experiments require two rounds of data preprocessing. First, for datasets with inconsistent topologies and varying facial poses as raw data, such as BU-3DFE and Headspace, we selected one facial scan as a template. We then conducted a Procrustes analysis based on the landmark data for each facial scan to align them. This allowed us to further use the selected intrinsic parameters to render the entire mesh dataset into an RGB-D dataset.

For the rendered datasets, we transformed them into valid input data for the FR system. These preprocessing steps include face detection and cropping. Note that we use the same parameters setting for MTCNN(Zhang et al., 2016b) as the face detectors for FaceNet and AdaFace. During the training process, for intermediate results generated, we used a face parser based on BiSeNet(Yu et al., 2018) to filter out irrelevant information such as background and neck regions.

For rendered 3D depth maps based on the FLAME topology, we first applied a pre-defined vertex mask to retain only the depth information of the facial region. After this step, we carried out preprocessing relevant to the FR system. The preprocessing pipeline comes from Led3D, which includes nose tip calibration, outliers removal, and depth normalization.

An example of data can be found in Figure 6.

### A.2   MASTER FACE ATTACK ANALYSIS

#### A.2.1   EVALUATION METRIC

Given $\boldsymbol{x}$ as the generated master face sample, from the context of our target, we typically employ False Matching Rate as the evaluation metric, which simply refers to:

$$\text{FMR} = \frac{\sum_{\boldsymbol{t} \in \mathbb{T}_h} m(\boldsymbol{x}, \boldsymbol{t}, \theta)}{\|\mathbb{T}_h\|}$$

In the above formula, the $m$ means the function for the FR system, either 2D or 3D, to determine whether the identity of the two inputs is the same. For the **joint FMR**, we use the match function $M$ to decide whether the two inputs are considered as the same person in both 2D and 3D scenarios:

$$\text{FMR}_{\text{joint}} = \frac{\sum_{\boldsymbol{t} \in \mathbb{T}_h} M(\boldsymbol{x}, \boldsymbol{t}, \theta_{\text{2d}}, \theta_{\text{3d}})}{\|\mathbb{T}_h\|}$$

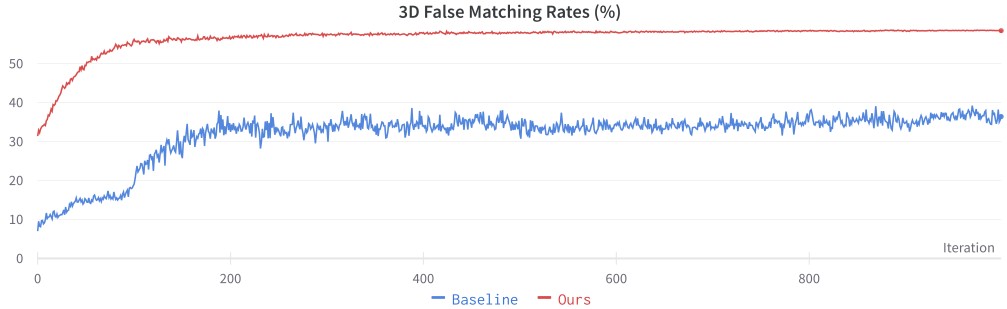

Figure 7: The training curve for the two master faces that were generated with baseline and our method, guided by the feedback of the 3D FR system only. Our methods show a better initialization and resulting FMR.

Apparently, the FMR is influenced by the choice of the training dataset and the performance of the FR systems selected. Due to variations in the assessment of different FR systems in individual research, there is currently no unified benchmark for evaluating the success rate of master face attacks. To the best of our knowledge, our research is the first attempt to simultaneously assess this success rate in both 2D and 3D systems.

Our analysis is primarily based on three key anchors. The first is the Equal Error Rate(EER) of the FR system we utilize, as it describes the performance of the FR systems, which essentially determines the FMR. The other anchor is the FMR of natural master faces. Natural faces refer to genuine faces in the dataset we want to evaluate. For example, if we intend to evaluate master face samples on the Texas3D dev dataset with a combination of AdaFace and Led3D as the FR systems, we first need to encode the face samples in this dataset using the FR system above, calculate their similarity scores, and identify a genuine master face, which is the face that has the highest number of matches with other enrolled face samples belong to different identities. Therefore, the calculation of the FMR for natural master faces is based on a white-box approach. However, when evaluating the master face samples we generate, because of inconsistent database distributions (BU-3DFE vs. Texas3D) and different FR systems used to train those master face samples (FaceNet vs. AdaFace, IResNet vs. Led3D), this success rate assessment can be seen as a black-box attack. The last anchor is also based on natural master faces, but the specific one that is computed with the same settings as how we train our master faces. Computing/Training from the same BU-3DFE training set with the FaceNet and IResNet FR systems, our mater faces generalize better than the natural master face, as shown in Table 4.

### A.2.2  ATTACK ON 3D FACE RECOGNITION SYSTEMS

This experimental design aims to validate our hypothesis that the master face generation method based on 3DMM can better learn from the shape information within the 3D facial dataset, resulting in a higher rate of false matching. In contrast, due to various factors such as optimization within the 2D latent variable space, unstable latent variable initialization, and errors in the 3D face reconstruction process, methods based on reconstruction have limited capabilities to preserve and utilize 3D shape information. In this experiment, we solely use the FMR computed from the 3D FR system as the objective function for the optimizer. The training curve obtained is shown in Figure 7, proving that the 3DMM-based method does outperform in learning crucial features for a 3D master face, resulting in higher 3D FMR.

### A.2.3  ATTACK ON 2D FACE RECOGNITION SYSTEMS

One of the criticisms against 3DMM is its issue with its blurring textures. To assess whether the method's 2D FMR gets affected, we solely utilized feedback from a 2D FR system to optimize the master face. This was designed to compare the final 2D FMRs between the 3DMM-based method and the reconstruction-based baseline. Since FaceNet performs exceptionally well, to avoid having the CMA-ES optimizer fail due to initially close zero FMR, we use a relatively low start point and

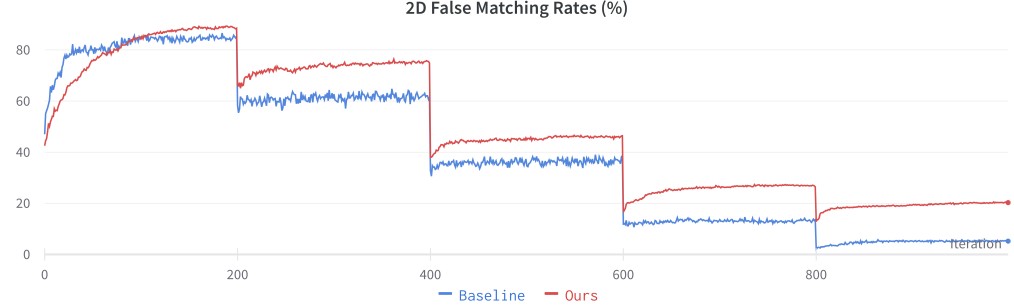

Figure 8: The training curve for the two master faces that were generated with baseline and our method, guided by the feedback of the 2D FR system only. Our method shows better robustness when the threshold decreases.

gradually increase FaceNet's matching threshold every 200 iterations. We found that the 3DMM-based method also outperformed the baseline method in terms of 2D FMRs, as shown in Figure 8. We hypothesize that 2D face recognition results are influenced by pose and expression. In our training dataset, all facial data maintained a frontal pose, which aligns with the fixed pose parameters of our 3DMM. In contrast, in StyleGAN, facial pose and expression are uncontrollable, which may have a negative impact on the final 2D error matching rates.

### A.2.4 ATTACK ON 2D AND 3D FACE RECOGNITION SYSTEMS CONCURRENTLY

The additional result of the comprehensive experiment settings depicted in Figure 3 is noted in the following Table 3.

We conduct evaluations over combinations of four 2D and 3D FR systems pairs (listed in the first column of Table 3) with three 3D facial datasets (from third to fifth columns for each sub-table), simulating in total 12 master face attack scenarios, including one white-box attack, two black-box attacks, and nine gray-box attack cases, colored from white to dark gray. Each row refers to the **2D, 3D, and joint FMRs** against the corresponding FR pair and dataset, computed with different strategies.

**Avg** and **Best** are computed from the natural faces belonging to the corresponding dataset, which are computed in **white-box** situations. They are employed as references to evaluate if our **Single, Greedy, and Morph** results can surpass the natural best result in white-box.

The **third to fourth rows** of each setting are respective evaluation results for a single master face instance and a set consisting of three master faces generated greedily via a reconstruction-based baseline. The **fifth to sixth rows** note the master faces generated with our 3DMM-based method instead. The final **Morph** row shows our key results, computed with the combination of those three master faces generated in greedy mechanism and their morphs in between, in total thirty samples for attack.

Another important clarification is, from **second to final** rows of each setting, the 2D and 3D FMRs are noted for the face instance with the best joint FMR. For example, in the white-box attack scenario, where we attack the BU-3DFE dev set with FR systems FaceNet and IResNet, our master faces generated with the strategy **Morph** have it has a high 3D FMR of 51.80%, but the 2D FMR 4.40% becomes its bottleneck. Our method is superior for the leverage of morphing, which significantly improves the 2D FMRs, hence leading to increasing joint FMRs.

**White-box Attack**   As indicated by the results of white-box attacks from the top left corner of Table 3, it is evident that a single master face generated by our method possesses a much higher level of joint false matching than the average of the original dataset. However, as we discussed in the experimental section, due to the discrepancies in the distributions of the two FR systems, the actual cluster of master faces that can be recognized by both 2D and 3D simultaneously may be

small and sparse. Therefore, a single master face attack may fail, even in the case of a white-box attack.

To increase the generalizability and effectiveness of the master face, we employ a greedy mode to generate three faces and take the union of the individuals they successfully match with (excluding duplicate matches) as the final result to estimate the potential master face clusters in the test set. Taking the BU-3DFE dev dataset as an example, the three faces achieve matching rates of 0.8%, 2.0%, and 0% respectively (as mentioned earlier, single master faces may have a chance of failure), and their matched individuals do not overlap, resulting in a 2.8% success rate.

Furthermore, we discovered that the morphing operation significantly increases the risk of master face attacks. First, morphing two existing master face samples is a simple and rapid process that does not require retraining and can generate a large number of new master faces quickly. Secondly, the morphing operation effectively establishes a linear interpolation in the latent variable space, and sliding along these latent codes generates new master faces that effectively cover more enrolled user templates. With morphing involved in the attack, we further increased the FMR from 2.8% to 4.4%.

**Black-box Attack**   The scenario of black-box attacks is presented in the lower right corner of Table 3. We employed two FR systems, AdaFace and Led3D, which were not used in training the master faces, and attempted to attack the test sets with different distributions of the BU-3DFE dataset, namely Headspace and Texas3D datasets. From the Equal Error Rates Table 1, AdaFace and Led3D indeed have higher EER on Texas3D compared to Headspace with the thresholds we set. However, our method did not succeed on Texas3D but achieved results of 0.2% and 0.4% on Headspace's dev and eval sets, respectively, demonstrating the possibility of success in black-box attacks.

We analyze that this might be due to our preprocessing procedure, as mentioned in Section A.1. We used the same preprocessing pipeline to render and preprocess the mesh data of Headspace and BU-3DFE. Hence, even though their original mesh data and topologies are different, the shared preprocessing process creates some commonalities in the final RGB-D datasets, enabling our master face attack to succeed. This demonstrates a possibility that even in real-world attack scenarios where the structure of the target FR system is unknown, there is still a possibility of success in using master faces for brute-force attacks if attackers hold a set of master faces trained on a sufficiently diverse facial dataset with different modes.

**Gray-box Attack**   Gray-box attacks can occur in various scenarios, such as when the database distribution is known while the architecture of the FR systems is (partially) unknown, or vice versa. In the case where the database distribution is known, as seen in the two columns for BU-3DFE dev and eval, our method exhibits excellent performance. Even in the last row where both FR systems (AdaFace and Led3D) are unknown, we still achieve FMRs close to, or even higher than, those of natural master faces. In scenarios where the FR systems are known but the database is unknown, our method continues to outperform natural master faces on Headspace. The performance decreases when the architectures of the FR systems are partially unknown, but we can still effectively launch attacks on the unknown database, like Headspace.

However, on Texas3D, our gray-box attack still failed. Combining the analysis from the previous section on black-box attacks, we speculate that even though both the FR system and database distribution have a strong impact on the final FMR, the database distribution may have a greater weight in determining the attack success rates. The reason might be that mainstream deep learning-based FR systems are mostly built upon convolutional neural networks and classical backbones, like ResNet for instance, which could lead to some commonalities in the facial features that they learned.

**Natural Master Face Attack**   As our master face is trained upon BU-3DFE trainset with FaceNet and IResNet, we conducted an additional experiment to measure the performance of the natural master face within this setting. To this end, we use this specific computed natural master face to perform the same attack settings as in Table 3, resulting in the evaluation results in Table 4. We found that the natural master face over the BU-3DFE trainset fails to generalize in most of the cases. In contrast, our synthesized master face, generated from the same dataset, shows significantly greater generalizability, which further proves the attacking ability of our methods.

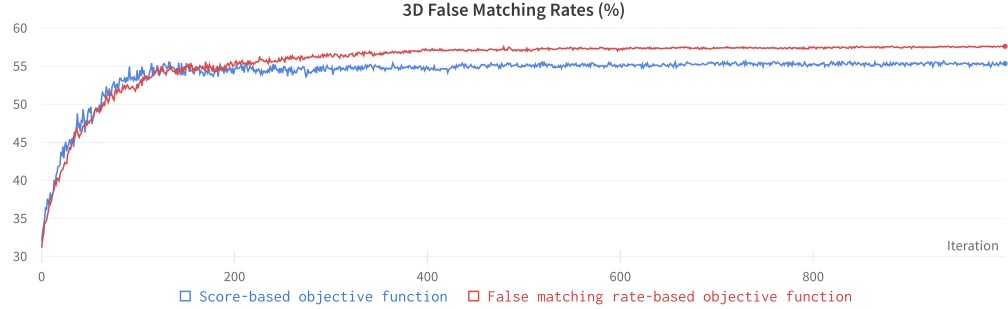

Figure 9: The training curve for the two master faces that were generated with baseline and our method, guided by the feedback of the 3D FR system only. The two master faces are both trained with our 3DMM-based method, starting with the zero vector as the shape vector. As observed, The trend of the 3D FMR is similar at the start of the optimization. After 1,000 iterations, the difference between the FMRs of these two master faces is 2.3%, showing that a score-based objective function, while not the optimal choice, still delivers respectable performance.

### A.3 ABLATION STUDY ON MASTER FACE GENERATION

#### A.3.1 OBJECTIVE FUNCTION SELECTION FOR CMA-ES SOLVER

As mentioned in Section 3, when the CMA-ES solver samples and provides possible candidate answers, we need to return the fitness scores corresponding to these answers to assist CMA-ES in further optimizing these latent variables. A good score function essentially plays a decisive role in the efficiency of optimization. Previous research on master faces has proposed two approaches based on similarity scores and false matching rates. We provide detailed reasons for choosing the latter in Section 3. As shown in Figure 9, when we optimize with a single-modal FR system, both objective functions yield similar results and efficiency. However, in the case of cross-modal optimization, using a score-based objective function causes the optimizer to focus on improving individual performance while ignoring the need to find a "cross-modal space." As a result, the FMR of the master face generated by the score-based function in 10 is much lower (nearly 0%) than the FMR-based function. The resulting shape images are summarized in Figure 12 for reference.

#### A.3.2 3D MORPHABLE FACE MODEL REGULARIZATION

One crucial point to note in the implementation is that 3DMM assumes the distribution of its parameters, assuming they follow a Gaussian distribution with a mean of zero. This assumption is violated during the optimization process of the CMA-ES solver, and the objective function we have chosen leads the optimizer to focus only on improving the false matching rate, without regard for whether the generated shapes remain anatomically plausible. To address this issue, we introduce a regularization term into the objective function to penalize shape codes that deviate too far from the zero vector. With this regularization term, the objective function used in our implementation can be formulated as:

$$L = 1 - \text{FMR}_{\text{joint}} + w\|\boldsymbol{\beta}\|_2$$

However, this regularization term to some extent limits the ability of the CMA-ES solver to optimize shape variables, as shown in Figure 11. Therefore, choosing an appropriate weight is important to balance between a high FMR and an anatomically plausible shape.

We put the shape images of two master faces generated with the same settings except for the weight for the regularization term in Figure 12.

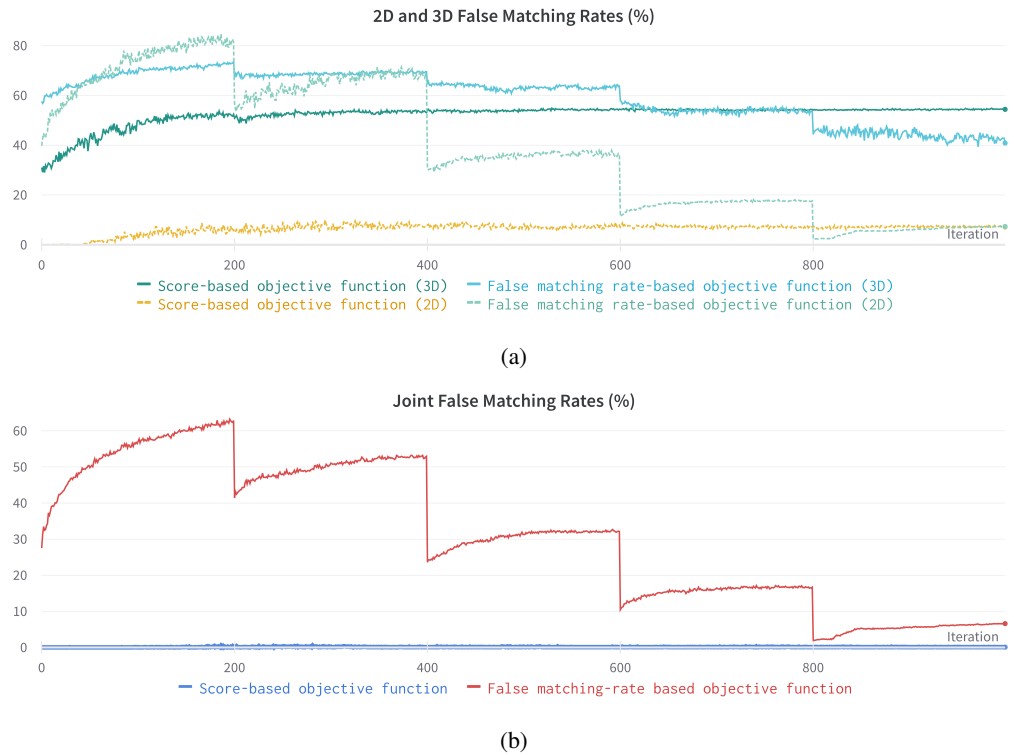

(a)

(b)

Figure 10: The training curve for the two master faces that were generated with baseline and our method, guided by the feedback of 2D and 3D FR systems together, aims to jointly attack these two cross-modal systems. Note that an FMR decrease for every 200 iterations is observed because we employ a threshold-increasing mechanism for the FMR-based method. As shown in Figure 10a, these two different objective functions in the end achieve similar FMR in 2D. For 3D FMR, the score-based function even outperforms. However, Figure 10b shows that the score-based function is failing to jointly attack the 2D and 3D systems. After 1,000 iterations, the FMR-based function has an FMR of 6.6%, while the score-based function holds only 0.06%.

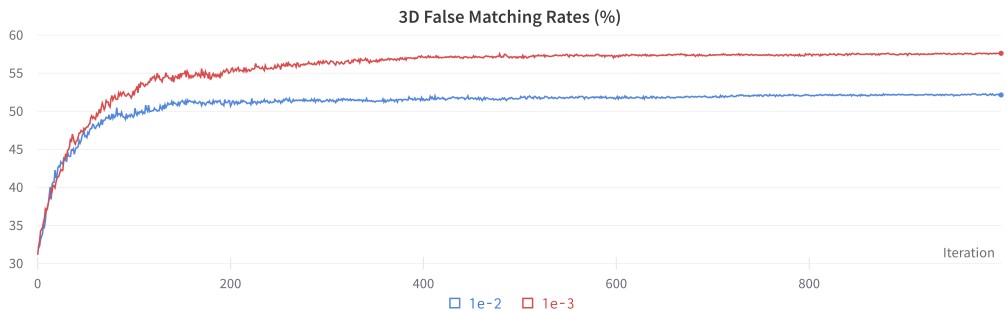

Figure 11: The training curve for the two master faces that were generated with baseline and our method, guided by the feedback of the 3D FR system only. It is evident that the larger regularization term limited the ability to further craft the shape code, resulting in lower 3D FMR.

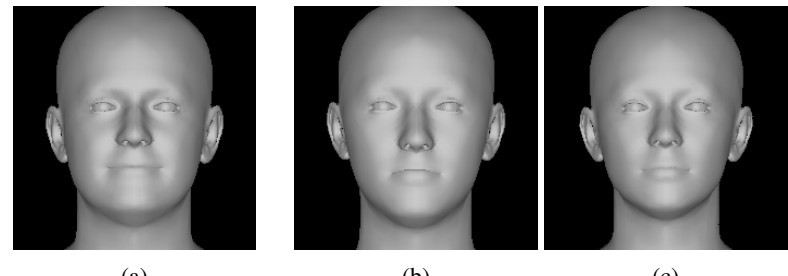

| (a) | (b) | (c) | (d) |

Figure 12: Shape images generated with different settings for reference. Figure 12a is from an initialized face with zero vector as shape code. Figure 12c is from the master face generated with the FMR-based objective function and a larger regularization term 1e-2. Figure 12b instead comes from the master face generated with the score-based objective function and a smaller regularization term 1e-3. And Figure 12d is from the master face with the best 3D FMR, generated with the FMR-based objective function and regularization term 1e-3.

### A.4 LIMITATION AND FUTURE WORK

Our 3DMM-based method for 3D master face generation still has below limitations: 1) Most 3DMM models have low dimensional albedo space, therefore cannot generate high-fidelity 2D faces that would convincingly deceive human eyes. This fact suggests that if 2D FMR can increase with the improvement of the albedo space, there may still be room for enhancement in joint FMR. 2) The LVE algorithm is less efficient, as it can optimize only one latent vector at a time. 3) Black-box master face attacks do not succeed when the distribution of the training dataset is dissimilar to that of the attack dataset.

For our future work, as the white-box and gray-box attack results still reveal the significant threats from master face attacks, especially those enhanced through morphing, we will explore potential countermeasures to bridge the gap in this field, as described in Section 5. Attempting to enhance the quality of 2D master faces generated by 3DMM or utilizing the differentiable properties of 3DMM to learn possible distributions of master faces, rather than individual latent vectors, is also a meaningful direction, as both attacks and defenses are crucial for the development in the field of security.

Table 3: Results for master face attack success rates simulated with different settings as depicted in Figure 3, divided into two sub-tables. Details illustration noted in Section A.2.

(a)

| FRs | Strategy | BU-3DFE dev (%) | | | BU-3DFE eval (%) | | | Headspace dev(%) | | |
|---|---|---|---|---|---|---|---|---|---|---|
| | | 2D | 3D | Joint | 2D | 3D | Joint | 2D | 3D | Joint |
| FaceNet IResNet | Avg | 1.09 | 9.06 | 0.01 | 1.39 | 13.99 | 0.35 | 3.89 | 3.56 | 0.34 |
| | **Best** | **1.20** | **31.60** | **0.80** | **10.60** | **34.40** | **7.40** | **17.56** | **11.78** | **4.19** |
| | Single | 0.00 | 6.80 | 0.00 | 3.20 | 5.20 | 0.00 | 0.20 | 0.20 | 0.00 |
| | Greedy | 0.20 | 23.80 | 0.00 | 3.20 | 28.60 | 0.00 | 7.78 | 1.40 | 0.00 |
| | Single | 0.80 | 40.00 | 0.80 | 4.20 | 56.60 | 4.20 | 5.99 | 6.79 | 1.00 |
| | Greedy | 3.00 | 48.40 | 2.80 | 15.40 | 64.60 | 14.00 | 15.97 | 16.37 | 2.59 |
| | *Morph* | *4.40* | *51.80* | *4.40* | *19.60* | *67.00* | *19.20* | *20.96* | *22.75* | *4.59* |
| FaceNet Led3D | Avg | 1.09 | 11.74 | 0.06 | 1.39 | 22.74 | 0.84 | 3.89 | 2.58 | 0.27 |
| | **Best** | **5.20** | **2.20** | **2.20** | **10.60** | **46.80** | **9.40** | **17.56** | **10.78** | **4.19** |
| | Single | 0.00 | 6.80 | 0.00 | 3.20 | 0.80 | 0.00 | 0.20 | 0.00 | 0.00 |
| | Greedy | 0.20 | 15.20 | 0.00 | 3.20 | 14.20 | 0.00 | 7.78 | 0.00 | 0.00 |
| | Single | 0.80 | 35.80 | 0.60 | 4.20 | 46.80 | 4.00 | 5.99 | 6.99 | 0.40 |
| | Greedy | 3.00 | 49.80 | 2.00 | 15.40 | 53.40 | 11.60 | 15.97 | 10.78 | 0.40 |
| | *Morph* | *4.40* | *55.40* | *4.20* | *19.60* | *60.60* | *18.20* | *20.96* | *13.77* | *2.20* |
| AdaFace IResNet | Avg | 9.88 | 9.06 | 1.91 | 9.96 | 13.99 | 3.04 | 3.39 | 3.56 | 0.31 |
| | **Best** | **36.40** | **40.40** | **16.60** | **31.60** | **47.80** | **22.00** | **18.56** | **13.97** | **4.99** |
| | Single | 0.60 | 6.80 | 0.00 | 4.20 | 5.20 | 0.00 | 0.00 | 0.20 | 0.00 |
| | Greedy | 2.60 | 23.80 | 0.00 | 6.20 | 28.60 | 1.00 | 0.00 | 1.40 | 0.00 |
| | Single | 5.20 | 40.00 | 5.20 | 4.80 | 56.60 | 4.80 | 0.00 | 6.79 | 0.00 |
| | Greedy | 8.40 | 48.40 | 7.00 | 8.20 | 64.60 | 7.40 | 0.40 | 16.37 | 0.00 |
| | *Morph* | *19.40* | *51.80* | *15.00* | *25.00* | *67.00* | *22.60* | *1.80* | *22.75* | *0.60* |
| AdaFace Led3D | Avg | 9.88 | 11.74 | 3.08 | 9.96 | 22.74 | 4.51 | 3.39 | 2.58 | 0.25 |
| | **Best** | **26.60** | **51.20** | **20.20** | **34.20** | **48.80** | **25.20** | **14.97** | **7.98** | **3.39** |
| | Single | 0.60 | 6.80 | 0.00 | 4.20 | 0.80 | 0.00 | 0.00 | 0.00 | 0.00 |
| | Greedy | 2.60 | 15.20 | 0.00 | 6.20 | 14.20 | 0.00 | 0.00 | 0.00 | 0.00 |
| | Single | 5.20 | 35.80 | 4.60 | 4.80 | 46.80 | 3.80 | 0.00 | 6.99 | 0.00 |
| | Greedy | 8.40 | 49.80 | 6.80 | 8.20 | 53.40 | 6.80 | 0.40 | 10.78 | 0.00 |
| | *Morph* | *19.40* | *55.40* | *17.00* | *25.00* | *60.60* | *20.60* | *1.80* | *13.77* | *0.40* |

(b)

| FRs | Strategy | Headspace eval (%) | | | Texas3d dev (%) | | | Texas3d eval (%) | | |
|---|---|---|---|---|---|---|---|---|---|---|
| | | 2D | 3D | Joint | 2D | 3D | Joint | 2D | 3D | Joint |
| FaceNet IResNet | Avg | 3.48 | 2.90 | 0.31 | 0.08 | 4.31 | 0.01 | 0.17 | 2.80 | 0.01 |
| | **Best** | **9.18** | **11.18** | **3.79** | **3.24** | **23.73** | **1.69** | **6.60** | **9.40** | **2.60** |
| | Single | 0.20 | 0.00 | 0.00 | 0.00 | 0.62 | 0.00 | 0.00 | 1.80 | 0.00 |
| | Greedy | 6.99 | 1.60 | 0.20 | 0.00 | 0.62 | 0.00 | 0.00 | 1.80 | 0.00 |
| | Single | 5.59 | 4.39 | 0.60 | 0.00 | 0.00 | 0.00 | 0.00 | 0.00 | 0.00 |
| | Greedy | 14.17 | 13.17 | 1.20 | 0.00 | 0.46 | 0.00 | 0.00 | 4.40 | 0.00 |
| | *Morph* | *20.76* | *18.76* | *4.19* | *0.00* | *0.46* | *0.00* | *0.00* | *4.40* | *0.00* |
| FaceNet Led3D | Avg | 3.48 | 2.06 | 0.20 | 0.08 | 3.81 | 0.05 | 0.17 | 12.25 | 0.05 |
| | **Best** | **11.18** | **11.18** | **2.40** | **8.32** | **20.18** | **7.55** | **11.40** | **8.00** | **5.00** |
| | Single | 0.20 | 0.00 | 0.00 | 0.00 | 0.00 | 0.00 | 0.00 | 0.00 | 0.00 |
| | Greedy | 6.99 | 0.00 | 0.00 | 0.00 | 0.00 | 0.00 | 0.00 | 0.00 | 0.00 |
| | Single | 5.59 | 6.99 | 0.40 | 0.00 | 0.00 | 0.00 | 0.00 | 0.00 | 0.00 |
| | Greedy | 14.17 | 10.18 | 0.60 | 0.00 | 0.62 | 0.00 | 0.00 | 0.00 | 0.00 |
| | *Morph* | *20.76* | *12.18* | *2.20* | *0.00* | *0.62* | *0.00* | *0.00* | *0.00* | *0.00* |
| AdaFace IResNet | Avg | 3.75 | 2.90 | 0.30 | 6.18 | 4.31 | 0.49 | 6.40 | 2.80 | 0.31 |
| | **Best** | **16.97** | **11.18** | **4.79** | **31.28** | **18.34** | **9.71** | **22.00** | **9.40** | **4.60** |
| | Single | 0.00 | 0.00 | 0.00 | 8.94 | 0.62 | 0.00 | 4.20 | 1.80 | 0.00 |
| | Greedy | 0.00 | 1.60 | 0.00 | 9.86 | 0.62 | 0.00 | 4.60 | 1.80 | 0.00 |
| | Single | 0.00 | 4.39 | 0.00 | 0.31 | 0.00 | 0.00 | 0.20 | 0.00 | 0.00 |
| | Greedy | 0.00 | 13.17 | 0.00 | 1.69 | 0.46 | 0.00 | 5.80 | 4.40 | 0.40 |
| | *Morph* | *1.60* | *18.76* | *0.20* | *4.01* | *0.46* | *0.00* | *18.20* | *4.40* | *0.60* |
| AdaFace Led3D | Avg | 3.75 | 2.06 | 0.18 | 6.18 | 3.81 | 0.68 | 6.40 | 12.25 | 1.50 |
| | **Best** | **23.75** | **10.98** | **4.39** | **28.51** | **20.18** | **14.79** | **27.00** | **28.20** | **13.60** |
| | Single | 0.00 | 0.00 | 0.00 | 8.94 | 0.00 | 0.00 | 4.20 | 0.00 | 0.00 |
| | Greedy | 0.00 | 0.00 | 0.00 | 9.86 | 0.00 | 0.00 | 4.60 | 0.00 | 0.00 |
| | Single | 0.00 | 6.99 | 0.00 | 0.31 | 0.00 | 0.00 | 0.20 | 0.00 | 0.00 |
| | Greedy | 0.00 | 10.18 | 0.00 | 1.69 | 0.62 | 0.00 | 5.80 | 0.00 | 0.00 |
| | *Morph* | *1.60* | *12.18* | *0.20* | *4.01* | *0.62* | *0.00* | *18.20* | *0.00* | *0.00* |

Table 4: Results for using a selected natural master face to attack the 12 settings as demonstrated in Figure 3. This natural master face is computed upon the BU-3DFE training set, the employed FR systems are FaceNet and IResNet. Its computation setting is exactly the same as our master face generation setting. The result of its FMR for each attacking setting is shown in the column **Natural**. We put our best results with the morphing strategy in the column **Morph** for comparison.

(a)

| FRs / Strategy | | Avg | Best | Natural | Morph | Avg | Best | Natural | Morph |
|---|---|---|---|---|---|---|---|---|---|
| | | | FaceNet IResNet | | | | FaceNet Led3D | | |
| BU-3DFE dev (%) | 2D | 1.09 | 1.20 | **0.00** | *4.40* | 1.09 | 5.20 | **0.00** | *4.40* |
| | 3D | 9.06 | 31.60 | **39.60** | *51.80* | 11.74 | 2.20 | **38.20** | *55.40* |
| | Joint | 0.01 | 0.80 | **0.00** | *4.40* | 0.06 | 2.20 | **0.00** | *4.20* |
| BU-3DFE eval (%) | 2D | 1.39 | 10.60 | **0.00** | *19.60* | 1.39 | 10.60 | **0.00** | *19.60* |
| | 3D | 13.99 | 34.40 | **46.20** | *67.00* | 22.74 | 46.80 | **39.00** | *60.60* |
| | Joint | 0.35 | 7.40 | **0.00** | *19.20* | 0.84 | 9.40 | **0.00** | *18.20* |
| Headsapce dev (%) | 2D | 3.89 | 17.56 | **0.20** | *20.96* | 3.89 | 17.56 | **0.20** | *20.96* |
| | 3D | 3.56 | 11.78 | **3.79** | *22.75* | 2.58 | 10.78 | **1.60** | *13.77* |
| | Joint | 0.34 | 4.19 | **0.00** | *4.59* | 0.27 | 4.19 | **0.00** | *2.20* |
| Headspace eval (%) | 2D | 3.48 | 9.18 | **0.80** | *20.76* | 3.48 | 11.18 | **0.80** | *20.76* |
| | 3D | 2.90 | 11.18 | **2.40** | *18.76* | 2.06 | 11.18 | **1.40** | *12.18* |
| | Joint | 0.31 | 3.79 | **0.00** | *4.19* | 0.20 | 2.40 | **0.00** | *2.20* |
| Texas3D dev (%) | 2D | 0.08 | 3.24 | **0.00** | *0.00* | 0.08 | 8.32 | **0.00** | *0.00* |
| | 3D | 4.31 | 23.73 | **0.00** | *0.46* | 3.81 | 20.18 | **0.00** | *0.62* |
| | Joint | 0.01 | 1.69 | **0.00** | *0.00* | 0.05 | 7.55 | **0.00** | *0.00* |
| Texas3D eval (%) | 2D | 0.17 | 6.60 | **0.00** | *0.00* | 0.17 | 11.40 | **0.00** | *0.00* |
| | 3D | 2.80 | 9.40 | **0.00** | *4.40* | 12.25 | 8.00 | **0.00** | *0.00* |
| | Joint | 0.01 | 2.60 | **0.00** | *0.00* | 0.05 | 5.00 | **0.00** | *0.00* |

(b)

| FRs / Strategy | | Avg | Best | Natural | Morph | Avg | Best | Natural | Morph |
|---|---|---|---|---|---|---|---|---|---|
| | | | AdaFace IResNet | | | | AdaFace Led3D | | |
| BU-3DFE dev (%) | 2D | 9.88 | 36.40 | **18.00** | *19.40* | 9.88 | 26.60 | **18.00** | *19.40* |
| | 3D | 9.06 | 40.40 | **39.60** | *51.80* | 11.74 | 51.20 | **38.20** | *55.40* |
| | Joint | 1.91 | 16.60 | **9.80** | *15.00* | 3.08 | 20.20 | **12.20** | *17.00* |
| BU-3DFE eval (%) | 2D | 9.96 | 31.60 | **17.80** | *25.00* | 9.96 | 34.20 | **17.80** | *25.00* |
| | 3D | 13.99 | 47.80 | **46.20** | *67.00* | 22.74 | 48.80 | **39.00** | *60.60* |
| | Joint | 3.04 | 22.00 | **12.00** | *22.60* | 4.51 | 25.20 | **10.40** | *20.60* |
| Headsapce dev (%) | 2D | 3.39 | 18.56 | **0.00** | *1.80* | 3.39 | 14.97 | **0.00** | *1.80* |
| | 3D | 3.56 | 13.97 | **3.79** | *22.75* | 2.58 | 7.98 | **1.60** | *13.77* |
| | Joint | 0.31 | 4.99 | **0.00** | *0.60* | 0.25 | 3.39 | **0.00** | *0.40* |
| Headspace eval (%) | 2D | 3.75 | 16.97 | **1.00** | *1.60* | 3.75 | 23.75 | **1.00** | *1.60* |
| | 3D | 2.90 | 11.18 | **2.40** | *18.76* | 2.06 | 10.98 | **1.40** | *12.18* |
| | Joint | 0.30 | 4.79 | **0.00** | *0.20* | 0.18 | 4.39 | **0.00** | *0.20* |
| Texas3D dev (%) | 2D | 6.18 | 31.28 | **7.70** | *4.01* | 6.18 | 28.51 | **7.70** | *4.01* |
| | 3D | 4.31 | 18.34 | **0.00** | *0.46* | 3.81 | 20.18 | **0.00** | *0.62* |
| | Joint | 0.49 | 9.71 | **0.00** | *0.00* | 0.68 | 14.79 | **0.00** | *0.00* |
| Texas3D eval (%) | 2D | 6.40 | 22.00 | **6.00** | *18.20* | 6.40 | 27.00 | **6.00** | *18.20* |
| | 3D | 2.80 | 9.40 | **0.00** | *4.40* | 12.25 | 28.20 | **0.00** | *0.00* |
| | Joint | 0.31 | 4.60 | **0.00** | *0.60* | 1.50 | 13.60 | **0.00** | *0.00* |

