# OpenReview forum: "3D Morphable Master Face Generation: Towards Controllable Wolf Attacks against 2D and 3D Face Recognition Systems"
_ICLR.cc/2024/Conference — ICLR 2024 Conference Withdrawn Submission_

### Official Review · Reviewer_Tr8G · 2023-10-25

**Soundness:** 3 good
**Presentation:** 2 fair
**Contribution:** 2 fair
**Rating:** 3
**Confidence:** 5

**Summary:**

Summary: In this paper, the authors study the master attack on 3D face recognition systems. They use combined 2D and 3D face similarities as a loss functions, CMA-ES as an optimizer, 3DMM for controlling the face properties.

**Strengths:**

1.	The first several sections are well-presented.
2.	The algorithm is simple and straightforward.

**Weaknesses:**

1.	Some consider that master attack is a real problem. This problem is existing when we run the system in identification mode, 1-to many matching, which is used in low security application. This attack is much less effective in verification mode, 1-to-1 matching. If a system is run on identification model, collecting data from one particularly registered user is very possible. Thus, target attack can be performed.
2.	The proposed work is an integration of the existing models and tools. It affects it novelty.
3.	This work can be considered as an extension of Friedlander et al 2022, which also generates 2D and 3D master faces. The proposed work uses better components to achieve better performance. It further weakens the novelty.
4.	The presentation of the experimental session is not good enough. I read several times to try to understand the contents. The authors also use abbreviation defined in appendix, e.g., dev. The table numbers in the main text are not in order. Very significant effort is needed to improve the presentation.
5.	Most of the attacks are ineffective. Many zero FMR in Table 3. Most important, in black-box setting, the attack is totally ineffective.
6.	The current study is in digital domain but 3D face is called in physical domain.
7.	Furthermore, all the previous related works are published in the biometric conference and biometric journals. Submitting this paper to e.g. IEEE Trans. On Biometrics, Behaviour and Identity Science or other biometric conferences, may be more suitable.

**Questions:**

See the comments in weaknesses.

---

> ### Author Response · Authors · 2023-11-15
>
> Thank you for your detailed feedback and suggestions. We are willing to discuss your concerns and questions below.
>
> &nbsp;
>
> **Weakness 1** *Some consider that master attack is a real problem. This problem is existing when we run the system in identification mode, 1-to many matching, which is used in low security application. This attack is much less effective in verification mode, 1-to-1 matching. If a system is run on identification model, collecting data from one particularly registered user is very possible. Thus, target attack can be performed.*
>
> We believe that **target attack and master attack are two different types of attack**. As you pointed out, in verification mode, when the system knows the user's identity in advance, e.g. unlocking a smartphone, the master face is less effective while **still possible** because we can generate multiple master faces that refer to different clusters of "similar registered faces" that can hit the target. But identification mode is also widely used in many access control scenarios where the input identity is not known in advance. In this case, if we have the victim's image, we can certainly use a target attack. However, if we don't have this information, it's more effective to use Master Face Attack. **In practice, the 3D images of a victim are not as easy to collect as the 2D counterpart, so by using the 3D master face attack, the attacker can avoid trying to collect the 3D images of the victim.**
>
> &nbsp;
>
> **Weakness 2-3** *The proposed work is an integration of the existing models and tools. It affects it novelty, This work can be considered as an extension of Friedlander et al 2022, which also generates 2D and 3D master faces. The proposed work uses better components to achieve better performance. It further weakens the novelty.*
>
> We hope that our response below regarding our contribution addresses your concern about our novelty, a detailed summary can be found above as the response to Reviewer FSLJ.
> 1. **Our work explored the practicality of the master face attack to compromise up-to-date face authentication systems(2D+3D)** as modern face authentication systems no longer rely solely on 2D face recognition systems.
> 2. Compared to the previous work, our work proposed a method to **directly generate master faces in the 3D domain to better utilize the 3D information**. With 3DMM we **generate the first controllable and morphable master faces** with higher false matching rates.
> 3. Our work is **the first to evaluate the master face's generalizability in grey-box and black-box settings** to discuss the practicality of the 3D master face attack in a real-world scenario.
> 4. We revealed that **even though 3D FR is considered more robust than 2D FR, they share the same weakness of non-uniform feature space distribution, and thus are vulnerable to the master face attack**.
>
> &nbsp;
>
> **Weakness 4** *The presentation of the experimental session is not good enough. I read several times to try to understand the contents. The authors also use abbreviation defined in appendix, e.g., dev. The table numbers in the main text are not in order. Very significant effort is needed to improve the presentation.*
>
> Thank you for your feedback, we omitted some descriptions for notes that should be common knowledge. **May I ask for more details for "The table numbers in the main text are not in order"?** Your detailed comment would definitely help us to refine the paper.
>
> &nbsp;
>
> **Weakness 6** *The current study is in digital domain but 3D face is called in physical domain.*
>
> Thanks for pointing this out. **Applying a 3D master face attack in the digital domain is possible** because the 3D information from the physical domain can be collected by a physical sensor, such as a mobile device with a depth sensor, and converted to a digital format, such as RGB-D information, and manipulated by a virtual camera. However, **the 3D master face presentation attack is also possible by introducing 3D masks**[2][3]. Regarding the physical attack using the master face, we also mentioned that **our 3DMM-based master face can be used to improve the 2D master face presentation attack** because it is controllable to perform different expressions with different poses while maintaining identity, which remains a difficult task for the stylegan-generated 2D master face.
>
> &nbsp;
>
> Due to the limitation of characters, we are sorry for putting the response to the left weakness into another comment.

---

> ### Author Response · Authors · 2023-11-15
>
> Below are responses for Weakness 5 and 7.
>
> &nbsp;
>
> **Weakness 5** *Most of the attacks are ineffective. Many zero FMR in Table 3. Most important, in black-box setting, the attack is totally ineffective.*
>
> Thank you for your careful consideration of Table 3 in the Appendix. However, I would like to clarify the zero FMR in the table. First, the task is already challenging because the 2D and 3D FRs typically have different distributions, and we want to find faces that together match the 2D and 3D FRs and generalize these faces to unseen datasets or unknown FR systems. For this challenging task setting, we believe that the synthesised face can be considered efficient if it has a performance similar to or equal to the best natural master face, which can only be found in the white box case.
>
> Within Table 3, there are seven rows for each experimental setting, the third and fourth being the baseline that we re-implement. **For our methods, especially the bottom column where we use morph master faces to attack, which shows our best performance, there are far fewer zero FMRs**. As we show and analyse, in the grey box scenario, when the attack dataset distribution is known, our methods show high FMR. Other times, when the FR systems are (partially) known, the performance drops, but still works in most cases. As you pointed out, generalizing the master face found in the training set to the black box attack is difficult and ineffective, which is also reported in the 2D master face-related research[1]. However, **it is important to note that this does not mean that a master face attack is not a threat in a real-world scenario because the real-world attack is rarely a perfect black-box setting, but rather a grey-box setting**. For example, attackers can buy commercial 3D face data for training or analyse commercial FR architectures via teacher-student learning. Moreover, advanced deep learning-based FR architectures are somewhat similar, e.g. based on CNN architecture, human faces also share a similar topology, which makes the approximation much easier.
>
> &nbsp;
>
> **Weakness 7** *Furthermore, all the previous related works are published in the biometric conference and biometric journals. Submitting this paper to e.g. IEEE Trans. On Biometrics, Behaviour and Identity Science or other biometric conferences, may be more suitable.*
>
> Besides the work mentioned in this paper, there is also work related to the master attack published on WACV[4]. As we are analyzing the vulnerability of the deep learning-based face authentication system, we think the topic still fits the "societal considerations of representation learning, including fairness, security, privacy, and interpretability and explainability". However, we do thank you for the suggestion and the deep look into the related field.
>
> &nbsp;
>
> We hope that the answers to the weaknesses and questions will more or less address your concerns. Please do not hesitate to reply if you have any further questions that you would like to discuss.
>
> &nbsp;
>
> Reference:
>
> [1] Nguyen, Huy H., et al. "Master face attacks on face recognition systems." IEEE Transactions on Biometrics, Behavior, and Identity Science 4.3 (2022): 398-411.
>
> [2] https://www.idiap.ch/en/dataset/3dmad
>
> [3] Erdogmus, Nesli, and Sebastien Marcel. "Spoofing face recognition with 3D masks." IEEE Transactions on Information Forensics and Security, vol. 9, no. 7, pp. 1084-1097, 2014
>
> [4] Nguyen, Huy H., et al. "Analysis of Master Vein Attacks on Finger Vein Recognition Systems." Proceedings of the IEEE/CVF Winter Conference on Applications of Computer Vision. 2023.

---

### Official Review · Reviewer_ARdD · 2023-10-29

**Soundness:** 2 fair
**Presentation:** 1 poor
**Contribution:** 2 fair
**Rating:** 3
**Confidence:** 3

**Summary:**

This paper presents the vulnerability of biometric authentication systems to master face attacks, which utilize generative models to create facial samples that can match multiple registered user templates. The authors propose a systematic approach for generating 3D master faces that can compromise both 2D and 3D face recognition systems. They introduce a novel framework that uses a latent variable evolution (LVE) strategy with a 3D Morphable Face Model (3DMM) to generate these master faces. The authors demonstrate the significant threat posed by these 3D master faces through simulations in various attack scenarios. They also explore the application of facial reenactment and morphing in the generated samples to enhance the efficacy of the master face attack. The authors highlight the importance of defending against master face attacks and call for further research on strengthening face recognition systems.

**Strengths:**

-With the fast development of AIGC, the problem of master face attacks discussed in this paper will be an important issue for the face recognition system. Thus, the task of solving the master face attacks is meaningful.
-The problem of generating 3D master faces for 3D face recognition systems may be firstly studied by the authors.
-A latent variable evolution (LVE) strategy with a 3D Morphable Face Model (3DMM) is used to generate the 3D master faces, and the facial reenactment and morphing are further explored.
-Experiments with different databases, deep face models, and attack scenarios has been demonstrated.

**Weaknesses:**

-The problem of master face attacks for 2D FR systems has been studied by Nguyen et al. (2020), Shmelkin et al. (2021) and Nguyen et al. (2022). The authors extend this issue to 2D and 3D FR systems, and thus the novelty of the proposed method is limited;

-The presentations and the layout of the whole paper is poor, the authors should re-organize the whole paper. Many information is not complete to the reader. The main part of the paper should be closed to itself, including the method and experiments. But there are many information are introduced in the APPENDIX;

- The comparisons to the baseline is not faithful to the reader. It should be also compare the proposed method with the previous method of generating master face attacks for 2D FR.

**Questions:**

-The experimetal setting for different databases should be clearly presentated.

-Generally speaking, 3D face shapes generated by using the latent codes of 3DMM has inherent disadvantages of linearity, and looks very similar to each other (no identity information), thus I am wonder why the generated 3D face shapes can match many gallery faces with a pre-defined match threshold? Since we know that the threshold value of a given 3D face recognition system has been defined and fixed once the system has been used.

---

> ### Author Response · Authors · 2023-11-15
>
> Thank you for your detailed feedback and suggestions. We are willing to discuss your concerns and questions below.
>
> &nbsp;
>
> **Weakness 1** *The problem of master face attacks for 2D FR systems has been studied by Nguyen et al. (2020), Shmelkin et al. (2021) and Nguyen et al. (2022). The authors extend this issue to 2D and 3D FR systems, and thus the novelty of the proposed method is limited.*
>
> We believe that extending a 2D task to a 3D environment does not mean that there is no innovation and challenge involved. While today's face authentication rarely relies solely on 2D FR, our work tends to evaluate and expose the vulnerability of current face authentication systems to the master face attack and proved that simply extending the previous work to 3D had relatively poor results on grey-box and black-box attack scenario, while with our proposed method the master face attacks show threatening.
>
> The clarification of our contribution is summarized and attached below, a detailed summary can be found above as the response to Reviewer FSLJ.
>
> 1. **Our work explored the practicality of the master face attack to compromise up-to-date face authentication systems(2D+3D)** as modern face authentication systems no longer rely solely on 2D face recognition systems.
> 2. Compared to the previous work, our work proposed a method to **directly generate master faces in the 3D domain to better utilize the 3D information**. With 3DMM we **generate the first controllable and morphable master faces** with higher false matching rates.
> 3. Our work is **the first to evaluate the master face's generalizability in grey-box and black-box settings** to discuss the practicality of the 3D master face attack in a real-world scenario.
> 4. We revealed that **even though 3D FR is considered more robust than 2D FR, they share the same weakness of non-uniform feature space distribution, and thus are vulnerable to the master face attack**.
>
> &nbsp;
>
> **Weakness 2** *The presentations and the layout of the whole paper is poor, the authors should re-organize the whole paper. Many information is not complete to the reader. The main part of the paper should be closed to itself, including the method and experiments. But there are many information are introduced in the APPENDIX*
>
> We regret that, due to space constraints, we have had to focus on presenting our work as comprehensively as possible in the main article, with supplementary content in the appendix. The appendix may appear lengthy due to the inclusion of figures and tables, but they are all related to and discussed in the main article, so there shouldn't be any missing information. However, we are updating the manuscript for a better presentation.
>
> &nbsp;
>
> **Weakness 3** *The comparisons to the baseline is not faithful to the reader. It should be also compare the proposed method with the previous method of generating master face attacks for 2D FR.*
>
> We aim to compromise both 2D and 3D FR, so we compare to the baseline. The baseline we compare with first generates a 2D stylegan face and then uses a 3D reconstruction method to get the 3D face, while the previous work on 2D master face attack only generates a 2D stylegan face. If we do not include 3D FR guidance and 3D reconstruction in Baseline's procedure, it works the same as a 2D master face attack function. **In Appendix A.2, we proved that our method is still superior in this case, which is equivalent to a comparison with the previous 2D master face attack method**. We regret that we put this comparison in the Appendix to save the space for main paper.
>
> &nbsp;
>
> **Question 1** *The experimetal setting for different databases should be clearly presentated.*
>
> Thank you for pointing this out. We believe this comment is related to the W2 that you found the evaluation difficult to understand, we will claim more details of our experimental setting from Figure 3 to help understanding.
>
> &nbsp;
>
> The response for Question 2 is a bit lengthy, due to the limitation of characters, we are sorry for putting it into another comment.

---

> ### Author Response · Authors · 2023-11-15
>
> Below are the response for question 2.
>
> **Question 2** *Generally speaking, 3D face shapes generated by using the latent codes of 3DMM has inherent disadvantages of linearity, and looks very similar to each other (no identity information), thus I am wonder why the generated 3D face shapes can match many gallery faces with a pre-defined match threshold? Since we know that the threshold value of a given 3D face recognition system has been defined and fixed once the system has been used.*
>
> As you said, we use a fixed threshold calculated from the dev set of the 3D face dataset that achieves the lowest equal error rate for our face recognition system.
>
> As to why the 3D face shapes generated by 3DMM can match many gallery faces with this predefined threshold, **we believe the answer lies in the nature of the existence of a master face**, as we discuss in Section 4.2 and also demonstrated by Nguyen et al.[1] Specifically, due to the non-uniform distribution of deep learning-based FR systems, there exist dense clusters within the feature space. If a face is encoded in these dense clusters, it could possibly be matched with the other faces that are also in the same cluster. **Even if we do not use a synthesised face, but only consider the real face in the 3D dataset, we can find some natural faces that process this "master face" property, which we call "natural master face"**, as shown in column b of Figure 3. We find a shape and appearance code pair for the 3DMM to produce a 3D face shape that is close to or within such a cluster in feature space. As shown in Figure 4, the leftmost master face corresponds to the blue data point that falls within a cluster, while the rightmost master face corresponds to the red data point that falls within another one. Using the linearity of the 3DMM, **we could morph the two master faces to somehow bridge the two clusters and also match some other faces in between the two clusters**. We hope we have explained why 3DMM-generated faces can match many gallery faces, and why we say that the morphing ability of our master face further improves the false matching ability significantly.
>
> &nbsp;
>
> We hope that the answers to the weaknesses and questions will more or less address your concerns. Please do not hesitate to reply if you have any further questions that you would like to discuss.
>
> &nbsp;
>
> Reference:
>
> [1] Nguyen, Huy H., et al. "Master face attacks on face recognition systems." IEEE Transactions on Biometrics, Behavior, and Identity Science 4.3 (2022): 398-411.

---

### Official Review · Reviewer_SgrS · 2023-11-07

**Soundness:** 2 fair
**Presentation:** 3 good
**Contribution:** 2 fair
**Rating:** 3
**Confidence:** 4

**Summary:**

This paper proposes an approach for generating master faces that can compromise both 2D and 3D face recognition systems.3D master faces are generated using the Latent Variable Evolution (LVE) algorithm with the 3D Face Morphable Model (3DMM). Simultaneous master face attacks in both white-box, gray-box, and black-box scenarios are simulated in thee experiments showing the threat posed by these 3D master faces to mainstream face authentication systems. Face morphing and facial reenactment are also experimented on generated samples, enhancing the efficacy of master face attacks.

**Strengths:**

The main contributions are:
- a method is proposed that enhances the threat and usability of 3D master faces.
- an evaluation is performed across diverse 3D face datasets and various face recognition systems, encompassing
both white-box, gray-box, and black-box attack scenarios to simulate real-world master face attacks.
- it is shown how a controllable master face can enhance potential attacks through facial reenactment and morphing.

**Weaknesses:**

- The contribution of the paper in terms of learning solutions and its relevance to the conference seem a bit below the standard for ICLR
- the 3D face recognition methods referred in the related work section are quite old (there are no methods cited after 2019)
- the 3DMM use is not well detalied, investigated and discussed.
- experiements are not fully convincing

**Questions:**

- the used 3DMM it is not well explained. It is not clear which is the model used and its topology (from Figure 1 it seems Flame topology). It is the Flame model or the Basel Face Model? Did the authors consider non-linear 3DMM learned using deep neural networks? Which are the differences in using one 3DMM or a different one? Which are the data used to learn the 3DMM in the different experiments?
- 3DMM are not very good to reproduce fine shape details of the face. So one question here is how this impact in the proposed solution. Is it something desired this lck of detail or a more precise 3D face generation solution could be helpful?
- the 3D information is transformed to depth images. How does this solution affect the method? Would it be possible to work directly in the 3D domain?
- the BU-3DFE and Texas3D datasets are quite low-resolution with smooth surfaces. So, it can be this lack of details can couple better with the used 3DMM. Authors should use higher-resolution 3D datasets. I strongly suggest to use the FRGC v2.0 dataset.

**Details Of Ethics Concerns:**

This paper presents an approach for controllable 3D master face attacks to face recognition systems. So, though the purpose of this research is to highlight potential security risks of face recognition systems and encourage the development of more robust and secure biometric authentication systems, there could be abusive or illegal applications of the findings of this work.

---

> ### Author Response · Authors · 2023-11-15
>
> Thank you for your detailed feedback and suggestions. We are willing to discuss your concerns and questions below.
>
> &nbsp;
>
> **Weakness 1** *The contribution of the paper in terms of learning solutions and its relevance to the conference seem a bit below the standard for ICLR*
>
> The clarification of our contribution is summarized and attached below, a detailed summary can be found above as the response to Reviewer FSLJ.
> 1. **Our work explored the practicality of the master face attack to compromise up-to-date face authentication systems(2D+3D)** as modern face authentication systems no longer rely solely on 2D face recognition systems.
> 2. Compared to the previous work, our work proposed a method to **directly generate master faces in the 3D domain to better utilize the 3D information**. With 3DMM we **generate the first controllable and morphable master faces** with higher false matching rates.
> 3. Our work is **the first to evaluate the master face's generalizability in grey-box and black-box settings** to discuss the practicality of the 3D master face attack in a real-world scenario.
> 4. We revealed that **even though 3D FR is considered more robust than 2D FR, they share the same weakness of non-uniform feature space distribution, and thus are vulnerable to the master face attack**.
>
> Regarding the relevance to ICLR, since **deep learning-based FR system is one of the most widely used representation learning applications** today, and our work **proposes a potentially threatening attack method to bypass these representation learning systems in the real world**, as well as revealing the common weakness of 3D and 2D FR systems, we believe that this work still meets the "societal considerations of representation learning, including fairness, safety, privacy, and interpretability and explainability" topic of ICLR.
>
> &nbsp;
>
> **Weakness 2**  *The 3D face recognition methods referred in the related work section are quite old (there are no methods cited after 2019)*
>
> Thank you for pointing this out. We understand your concern that using outdated 3D FR may reduce the reliability of the result. However, as we mentioned in the related work section, although we benefit from industrial 3D FR in our daily life today, **the 3D FR research community is less thriving than 2D FR due to the scarcity of publicly available 3D face datasets for research purposes** and the nature of deep learning-based methods that require large amounts of training data. In our work, we use two 3D FRs. One is the Led3D, which was published at CVPR 2019 and is still the SOTA model among open-source related research, see [1], and the other is the IResNet backbone tuned with FaceScape 3D data, published at ICPR 2020, **which is also a reliable backbone widely used in many classification models today [2]**. Moreover, the aim of our work is to discuss and reveal the weaknesses of the 3D and 2D FR systems that make them vulnerable to the master face attack, rather than to propose the best model for the attack. Therefore, we believe that the use of these two methods won't affect the reliability of our results and conclusions.
>
> &nbsp;
>
> **Question 1** *The used 3DMM it is not well explained. It is not clear which is the model used and its topology (from Figure 1 it seems Flame topology). It is the Flame model or the Basel Face Model? Did the authors consider non-linear 3DMM learned using deep neural networks? Which are the differences in using one 3DMM or a different one? Which are the data used to learn the 3DMM in the different experiments?*
>
> In the Introduction and Method section, we explicitly state that we are using FLAME. As written in the Method section and in the overview Figure 1, **we do not learn a 3DMM from a 3D dataset, but rather use a FLAME model to generate a 3D face, and our goal is to learn the input shape and appearance codes as input to FLAME, these codes should generate the master faces that can be matched as much as possible to the registered samples in the FR system's gallery**. Our idea is that using 3DMM as a 3D master face generator is better than using a stylegan + 3D reconstruction method. Since the FLAME topology is also used in SOTA 3D face reconstruction methods such as DECA and MECA, we believe that its performance is sufficient to prove our idea. Regarding the nonlinear 3DMM, we appreciate your suggestion, and the greater representational power of the nonlinear 3DMM than the linear 3DMM may potentially improve our result and also provide a good insight for further work. **But for the scope of this work, using linear or non-linear 3DMM will not change the conclusion: the 3DMM-based method gives better 3D master face attack and allows controllability and morphs than the reconstruction-based method.**

---

> ### Author Response · Authors · 2023-11-15
>
> Below are responses for questions 2-4.
>
> &nbsp;
>
> **Question 3** *the 3D information is transformed to depth images. How does this solution affect the method? Would it be possible to work directly in the 3D domain?*
>
> Thank you for pointing this out.  **We transformed the real 3D face dataset and the synthesised face into depth images to match the input of the 3D FR systems we used**. Since the FLAME topology is fixed and the real 3D face dataset is pre-registered, we render the faces with a fixed viewpoint (i.e. the frontal view). As you may be concerned, **rendering the 3D face mesh on depth images can potentially lose viewpoint information and thus not fully exploit the existing 3D information. The accuracy of the rendering process is also important**. To mitigate this effect on the accuracy of the rendering process, **we use a pre-processing step before feeding the depth map into 3D FR with outlier removal and depth normalization**. Furthermore, **the effect of losing other viewpoint information is trivial in our context** because in most cases face authentication is done when the user turns the face in front of the camera, so the 3D information from other viewpoints is not that important. However, you are right that **it is also possible to work directly in the 3D domain if we switch to using a 3D FR based on a point cloud**. However, we believe that most 3D FR systems in everyday use are still depth map-based, as the depth sensor is more commercially available than the laser for point cloud retrieval.
>
> &nbsp;
>
> **Question 2 and 4** *3DMM are not very good to reproduce fine shape details of the face. So one question here is how this impact in the proposed solution. Is it something desired this lck of detail or a more precise 3D face generation solution could be helpful? the BU-3DFE and Texas3D datasets are quite low-resolution with smooth surfaces. So, it can be this lack of details can couple better with the used 3DMM. Authors should use higher-resolution 3D datasets. I strongly suggest to use the FRGC v2.0 dataset*
>
> Thank you for this thoughtful question. Putting these two questions together, we wonder if you mean that the proposed solution has a good result because the relatively poor quality of the 3DMM-generated faces is good for the 3D dataset we selected, which is also relatively low resolution. In other words, you are concerned that the 3DMM-based method may not work as well when the training set is of higher resolution, such as the FRGC v2.0 dataset. To clarify this, we first want to emphasise that we have compared our method with the baseline method, which is to reconstruct a stylegan-generated face image. We use DECA to re-implement the baseline, which **leads to a fair comparison because DECA also produces a FLAME topology face as output**. When the output is both in FLAME topology(same resolution), our method still outperforms the baseline.
>
> Our current experimental settings are better suited to a real-world scenario, as **most applications of 3D FR systems that we have encountered in everyday life are based on commercial 3D sensors, which typically have a lower resolution**. However, we appreciate your suggestion to include a higher resolution 3D face dataset to address more scenarios. Unfortunately for now as we currently have limited high-resolution datasets available (FaceScape is high resolution but the mesh lacks eye and mouth details), we may not be able to analyze the generalizability of the master face generated with high-resolution 3D face data.
>
> &nbsp;
>
> We assume that the above questions represent your detailed concerns of the Weakness 3 and 4. And we hope that the answers to the weaknesses and questions will more or less address your concerns. Please do not hesitate to reply if you have any further questions that you would like to discuss regarding 3DMM or experiments, .etc.
>
> &nbsp;
>
> Reference:
>
> [1] Li, Menghan, Bin Huang, and Guohui Tian. "A comprehensive survey on 3D face recognition methods." Engineering Applications of Artificial Intelligence 110 (2022): 104669.
>
> [2] https://insightface.ai/projects

---

### Official Review · Reviewer_FSLJ · 2023-11-10

**Soundness:** 3 good
**Presentation:** 3 good
**Contribution:** 2 fair
**Rating:** 5
**Confidence:** 4

**Summary:**

The authors aim at presenting a systematic approach for generating master faces that can compromise both 2D and 3D face recognition systems using the Latent Variable Evolution with the 3D Face Morphable Model.

**Strengths:**

The paper is well-written.
The motivation is clear.
The experimental setup is extensive.

**Weaknesses:**

Although the paper has a good potential, the paper does not provide sufficient novelty for publication. I was wondering if the authors could provide a summary of the major contributions of the paper.

**Questions:**

Please see above.

---

> ### Author Response · Authors · 2023-11-15
>
> Thank you very much for your feedback and the question.  Our contributions are summarized below.
>
> &nbsp;
>
> First, we want to discuss why studying 3D master face attacks is important. While previous work has focused on 2D master face attacks, **modern face authentication systems no longer rely solely on 2D face recognition systems**, which can be vulnerable to spoofing. Therefore, we need to study the 3D master face attack and investigate whether it could be a potential security issue for modern face authentication systems using 2D and 3D FR jointly.
>
> &nbsp;
>
> Next, we would like to clarify our contribution compared to the first study on 3D master face attacks proposed by Friedlander et al. (2022). **Their work was conducted and evaluated using Texas3D data, which is a white box attack scenario, and therefore does not provide further analysis for generalizing master face**. However, in real-world scenarios, attackers typically don't have full access to such detailed information. To fully assess the potential threat of 3D master face attacks, **we conducted extensive simulations in grey and black box attack scenarios.**
>
> &nbsp;
>
> In addition, we proposed our own methods using 3DMM to generate 3D master faces. While the baseline generates the faces in 2D space and reconstructs them later, we directly generate the face in 3D to better exploit the 3D information from the training set. Also, our proposed method **provides the first controllable and morphable master face that has not been explored in previous work**. We also found that these new properties make our 3D master face even more threatening than previous methods, **significantly improving the practicality of real-world master face attacks.**
>
> &nbsp;
>
> Combining the above contributions, we demonstrated that **even though 3D FR is considered more robust than 2D FR, they share the same weakness of non-uniform feature space distribution, and thus are vulnerable to the master face attack**. This has highlighted the societal consideration for the security of today's deep learning-based face authentication systems.
>
> &nbsp;
>
> We hope that our summary clarifies our contributions. Please do not hesitate to reply if you have any further questions that you would like to discuss.